# METATOOL BENCHMARK FOR LARGE LANGUAGE MODELS: DECIDING WHETHER TO USE TOOLS AND WHICH TO USE

**Yue Huang**[1*†]**, Jiawen Shi**[2]**, Yuan Li**[3]**, Chenrui Fan**[2]**, Siyuan Wu**[2]**, Qihui Zhang**[1†]
**Yixin Liu**[1]**, Pan Zhou**[2]**, Yao Wan**[2]**, Neil Zhenqiang Gong**[4]**, Lichao Sun**[1*]
Lehigh University[1]
Huazhong University of Science and Technology[2]
University of Cambridge[3]
Duke University[4]

## ABSTRACT

Large language models (LLMs) have garnered significant attention due to their impressive natural language processing (NLP) capabilities. Recently, many studies have focused on the tool utilization ability of LLMs. They primarily investigated how LLMs effectively collaborate with given specific tools. However, in scenarios where LLMs serve as intelligent agents, as seen in applications like AutoGPT and MetaGPT, LLMs are expected to engage in intricate decision-making processes that involve deciding whether to employ a tool and selecting the most suitable tool(s) from a collection of available tools to fulfill user requests. Therefore, in this paper, we introduce METATOOL, a benchmark designed to evaluate whether LLMs have tool usage awareness and can correctly choose tools. Specifically, we create a dataset called TOOLE within the benchmark. This dataset contains various types of user queries in the form of prompts that trigger LLMs to use tools, including both single-tool and multi-tool scenarios. Subsequently, we set the tasks for both tool usage awareness and tool selection. We define four subtasks from different perspectives in tool selection, including *tool selection with similar choices*, *tool selection in specific scenarios* , *tool selection with possible reliability issues*, and *multi-tool selection*. We conduct experiments involving eight popular LLMs and find that the majority of them still struggle to effectively select tools, highlighting the existing gaps between LLMs and genuine intelligent agents. However, through the error analysis, we found there is still significant room for improvement. Finally, we conclude with insights for tool developers – we strongly recommend that tool developers choose an appropriate rewrite model for generating new descriptions based on the downstream LLM the tool will apply to. Our TOOLE dataset is available at URL and code is in Github.

## 1 INTRODUCTION

Tool-empowered large language models (LLMs) (Qin et al., 2023a;b; Patil et al., 2023; Ruan et al., 2023; Cao et al., 2023; Zhou et al., 2023a) have recently attracted widespread attention. An important milestone for LLMs marching toward intelligent agents (Park et al., 2023; Li et al., 2023e) is the flexible use of tools (e.g., APIs (Qin et al., 2023b; Rapid, 2023) and plugins (OpenAI, 2023d)) to fulfill users' requirements. By utilizing tools, LLMs can obtain real-time data, such as getting the latest weather forecast (GPTStore, 2023); enhance interactions with users, like helping users book flight tickets (Deng et al., 2023); and better deal with uncertain questions by querying knowledge bases (Li et al., 2023c; Hu et al., 2023) or Internet (Lazaridou et al., 2022). Moreover, LLMs can also leverage specific tools to process multimodal information, thereby acquiring the same capabilities as multimodal models (Zhang et al., 2023; Yan et al., 2023; Yuan et al., 2023b). The capacity to use

---

[*]Lichao Sun and Yue Huang are co-corresponding authors: lis221@lehigh.edu, howiehwong@gmail.com
[†]Visiting Students at LAIR Lab, Lehigh University.

Table 1: Comparison of previous work and METATOOL.

| Dimension | APIBank (Li et al., 2023d) | GPT4Tool (Yang et al., 2023c) | APIBench (Patil et al., 2023) | ToolLLM (Qin et al., 2023b) | ToolBench (Xu et al., 2023) | ToolQA Zhuang et al. (2023) | MetaTool (Ours) |
|---|---|---|---|---|---|---|---|
| Evaluation Range | ①③④ | ③④ | ③④ | ③④ | ②③④ | ③④ | ①② |
| Number of Tasks | 1 | 1 | 1 | 1 | 1 | 1 | 4 |
| Reliability Test | ✗ | ✗ | ✔ | ✗ | ✗ | ✗ | ✔ |
| Multi-Tool Test | ✗ | ✗ | ✗ | ✔ | ✗ | ✔ | ✔ |
| Different Scenarios | ✗ | ✗ | ✗ | ✔ | ✗ | ✗ | ✔ |

tools enables LLMs to break through their own limitations, acquire external information, and thereby make more accurate and effective responses, providing users with better service.

Previous research has focused on how to enhance the ability of LLMs to use tools, including training models with instruction related to tool usage (Qin et al., 2023b; Tang et al., 2023; Schick et al., 2023), or augmenting the model's problem-solving capabilities for domain-specific tasks through external APIs (Yang et al., 2023c). A typical process of employing LLMs to use tools is illustrated in Figure 1. Initially, users input a question (i.e., query) that triggers the tool usage. Based on prior research (Yang et al., 2023c; Qin et al., 2023b), under the ReAct (Yao et al., 2022) prompt approach, the process of using tools can be divided into four stages: Firstly, LLMs consider whether to employ a tool (①) and if so, which tools to select (②). The tool selection process involves directly having LLMs choose from a provided tool list (Yang et al., 2023c) or selecting via a retriever (Qin et al., 2023b). Next, LLMs configure the users' input as tool parameters (③), then handle the results from the tool (④), and finally return the outcomes to the user.

With the emergence of more and more LLMs like open-source Llama2 (Touvron et al., 2023), Vicuna (Chiang et al., 2023), and closed-source ones like ChatGPT (OpenAI, 2023a) and GPT-4 (OpenAI, 2023b), designing a comprehensive benchmark to measure the tool-related capability of these models has become crucial. Current studies have proposed several benchmarks (Xu et al., 2023; Qin et al., 2023b; Li et al., 2023d) about tool us-

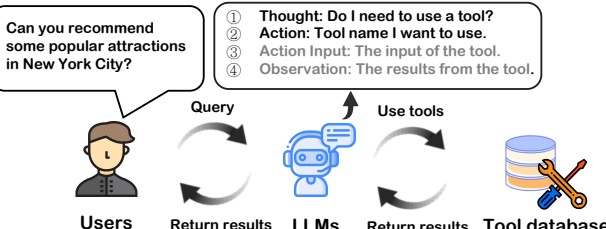

Figure 1: Tool usage pipeline of LLMs. METATOOL including awareness of tool usage (①) and tool selection (②).

age for LLMs, with the main contributions being limited to the stages ③ and ④. However, the awareness of tool usage (①) and tool selection (②) ability are also important for LLMs when they're acting as intelligent agents including AutoGPT (Significant-Gravitas, 2023), MetaGPT (geekan, 2023) and BabyAGI (babyagi, 2023), or in the multi-agent environment where LLMs need to use tools to solve collaborative tasks (Shen et al., 2023; Qian et al., 2023; Park et al., 2023; Cai et al., 2023). As a result, it is necessary to establish a benchmark to evaluate LLMs' tool usage consciousness and tool selection ability.

The difficulty in establishing such a benchmark is reflected in two aspects. The first one is the dataset: previous research proposed datasets (Qin et al., 2023b; Xu et al., 2023) lacked diverse user inputs, making it hard to cover various real-world scenarios. Additionally, there is an issue of overlapping in the dataset, meaning that a user's needs can be addressed by more than one tool, which makes it challenging to conduct evaluations since user inputs can correspond to multiple tools. The second aspect is the task setting: the benchmark should include different tasks to evaluate LLMs from different perspectives, such as reliability, the performance under different scenarios in daily life. To address these issues, we propose METATOOL, a benchmark designed to evaluate the awareness of tool usage and tool selection capability of LLMs. As demonstrated in Table 1, METATOOL distinguishes itself from previous research efforts and is structured into three primary components:

- **TOOLE dataset.** We introduce TOOLE, a comprehensive dataset that encompasses a wide range of 21,127 user queries, with both single-tool and multi-tool queries. Different from the previous single-method generation (Yang et al., 2023c; Qin et al., 2023b), these queries are generated using various prompting methods, including emotional generation, keyword generation, direct

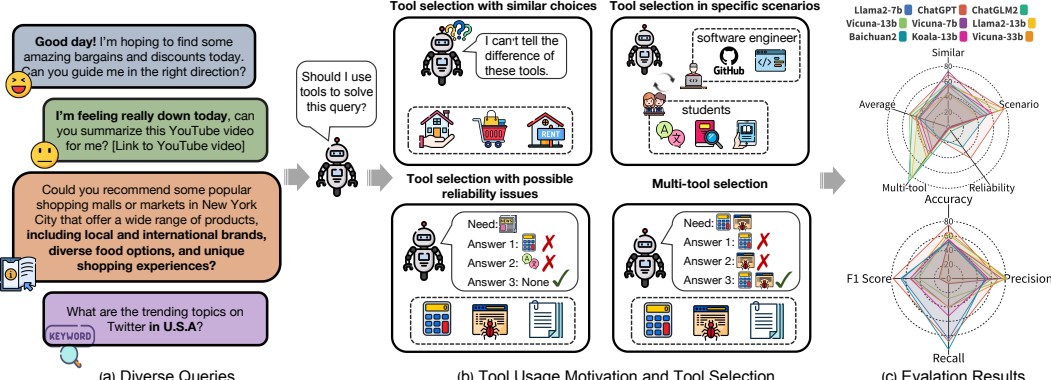

Figure 2: METATOOL benchmark architecture. It contains the dataset TOOOLE with diverse queries related to different tools (a), and based on it, we conduct the evaluation of the awareness of tool usage and tool selection (b) and finally obtain the results of eight prominent LLMs (c).

diverse generation, and detailed generation. Moreover, to address the challenge of overlapping tool functionality, we undertake tool merging and decomposition.

• **Evaluation on awareness of tool usage and tool selection.** We construct a test set to evaluate the awareness of tool usage based on TOOLE and existing instruction datasets. Moreover, we formulate four distinct tasks to evaluate the tool selection ability of LLMs. These tasks are thoughtfully designed to assess semantic comprehension, adaptability, reliability, and inferential capability, namely *tool selection with similar choices*, *tool selection in specific scenarios*, *tool selection with possible reliability issues*, and *multi-tool selection*.

• **Empirical analysis on results.** We rigorously evaluate the performance of eight well-known LLMs. We have observed that most LLMs struggle to recognize their capability boundaries and lack a good awareness of tool usage. Regarding tool selection, we find that while LLMs possess basic tool selection capabilities, the tool selection of most LLMs remains unreliable, with noticeable variations in performance across different daily scenarios. Moreover, the error analysis indicates there is still room for improvement in tool selection. Finally, by analysis the tool description, we gained two insights for tool developers.

## 2 METATOOL DESIGN

### 2.1 PRELIMINARY & REQUIRED ABILITIES

In this section, we first introduced the composition of the TOOLE dataset, outlining how we generated user queries related to tools. Subsequently, we explained how we set up the evaluation tasks, including tool usage awareness and tool selection. The evaluation mainly requires LLMs with the following properties and abilities: (1) Less hallucination and sycophancy. The awareness of tool usage can reflect the truthfulness about whether an LLM has a clear understanding of its capabilities (e.g., realizing its capability limitation about what problems it cannot solve well and using tools for assistance), thereby helping to mitigate issues of hallucination (Ji et al., 2023; Sun et al., 2024) and sycophancy (Wei et al., 2023). (2) Recommendation and retrieval. Moreover, existing research has tentatively explored the potential of LLMs in applications like LLM-based recommendation systems (e.g., tool recommendation for users) (Gao et al., 2023; Wang et al., 2023e; Dai et al., 2023). In LLM-as-agent scenarios, LLMs usually need to select the specific tool according to the text description (Park et al., 2023; Shen et al., 2023; Ruan et al., 2023), actually is a kind of information retrieval (Sun et al., 2023), making the ability of tool selection crucial. (3) Task-level abilities. In METATOOL, we set four tasks as shown in Table 12. Incorporating similar tools for selection (i.e., Task 1) requires a high-level semantic comprehension for LLMs, and tool selection in specific scenarios tests the flexibility of LLMs when using tools in different scenarios (e.g., finance (Wu et al., 2023a) and biomedical domain (Zhang et al., 2023; Wang et al., 2023c)). Task 3 aims to explore the internal hallucination and reliability extent of LLMs when using tools and Task 4 is designed to evaluate the inference ability (e.g., order of using multiple tools) (Creswell et al., 2022) of LLMs.

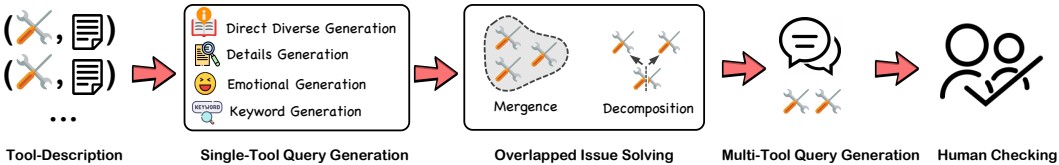

Figure 3: The process of dataset generation.

## 2.2 TOOLE DATASET

In this section, we introduce the TOOLE dataset with 21.1k diverse user queries related to tool usage. Each entry within the dataset comprises a user request (i.e., query) along with its corresponding tool name and tool description. These queries serve as triggers that prompt LLMs to utilize specific tools. The step-by-step process employed for generating the dataset is shown in Figure 3.

### 2.2.1 DATASET GENERATION

**Tool description.** Tool description is important for LLMs to use them (Hsieh et al., 2023). We retrieve tool names and descriptions from OpenAI's plugin list (OpenAI, 2023d). The reason for selecting Open AI plugins as the data source for our tools is that these tools have been installed in ChatGPT (OpenAI, 2023a) and GPT-4 (OpenAI, 2023b), and they have been widely used, making them more practical. We obtained names and descriptions for a total of 390 tools across different domains. We show more details about tool descriptions in Appendix A.5.

**Single-tool queries generation.** Next, we describe how we generated queries. Inspired by prior studies (Qin et al., 2023b; Yang et al., 2023c), our approach revolves around incorporating a tool's description into a prompt while implementing specific constraints to guide the generation of user queries by ChatGPT/GPT-4. We adopt four distinct techniques for query creation: direct diverse generation, emotional generation, keyword generation, and details generation. We show the data examples generated by different prompt ways in Table 8 of the Appendix A.5. (1) *Direct diverse generation.* We introduced conditional criteria within the prompt to encourage ChatGPT/GPT-4 to produce a variety of query types, encompassing distinct tones (such as requests or orders) and levels of detail. (2) *Emotional generation.* Building on prior research (Li et al., 2023a; Goodside, 2023), which highlights the influence of emotion within prompts on model performance, we augmented the prompt with constraints to guide ChatGPT in generating content in different emotions. Here we used four distinct emotions - happiness, excitement, anger, and depression. (3) *Keyword generation.* Direct generation occasionally fell short in capturing specific description details, such as tools limited to particular regions, so we devised the generation way through keywords. This way involved ChatGPT extracting keywords from the tool's description and then we incorporated both the extracted keyword and the tool's description within the prompt, tasking ChatGPT with generating queries focused on the given keyword. (4) *Details generation.* To add more details to the queries, we instructed ChatGPT to add details to augment the original queries generated by direct diverse generation methods.

**Overlapped issue.** Overlapped issue refers to *a query that can be solved by multiple tools*. If left unaddressed, this overlap could potentially influence the computation of final metrics. For instance, given a query $q$, the corresponding tool in our dataset is $t_a$, yet an alternate tool $t_b$ could also feasibly address the same query $q$. In a single-label scenario, the accuracy of tool selection becomes compromised. To address this, we merge the group of tools with similar functions as a single tool. Meanwhile, if a tool can function for multiple purposes across the groups of tools, the corresponding generated queries cannot be simply merged into any one of them. So decompositions are needed for the queries of these tools before merging. After decomposition and merging, each query in our dataset has only one ground truth label. The decomposition and merging operation follows three steps, and details about this can be found in Appendix A.1. We also show the efficiency of our operation in Appendix A.2 through the silhouette coefficient.

**Multi-tool queries generation.** Unlike single-tool queries, we generate multi-tool queries after addressing the overlapped issue because it is challenging to map the original labels to new labels in multi-label (i.e., multi-tool) situations. Here, we only consider queries related to two tools. We observe that if we obtain combinations of two tools by iterating through all the tools (i.e., $C_n^2$ iterations,

where $n$ is the size of the toolset), there would be many tool combinations that are not practical (i.e., rarely encountered in daily life, such as the combination of fortune-telling tools and currency exchange tools). Therefore, we select the top 15 most popular tools from the toolset, and for each pair of tools, we generate 5 queries. We determine the popularity of a tool based on the number of tools it is merged with, as shown in Appendix A.5. The multi-tool queries we generate can be divided into two types: The first category pertains to situations where tools are employed in parallel, indicating that the utilization of each tool operates independently of the others. The second category deals with cases where tools are used causally, signifying that the deployment of one tool may be contingent upon the outcomes of a preceding tool. Detailed prompt templates can be found in Appendix D.2. Similar to single-tool queries, we also manually verified multi-tool queries to ensure the combination of tools was reasonable and the query of the tool corresponded to the tool description.

**Human checking.** We conducted manual verification of all queries in TOOLE, including the removal of non-compliant queries and tools, as well as the handling of queries corresponding to special categories of tools. Detailed guidelines for human validation are provided in Appendix A.3.

## 2.3 TASK FORMULATION

We seek to address two research questions in this paper: (1) To what extent can LLMs be conscious of their limitations and ask for assistance from external tools? (2) How effectively can LLMs select the tools when they ask for assistance? To answer these questions, we design two tasks based on the TOOLE dataset to evaluate the capacity of LLMs regarding tool usage.

### 2.3.1 AWARENESS OF TOOL USAGE

In this part (i.e., *Thought* (①)), we aim to investigate the awareness of tool usage of LLMs; that is, whether LLMs can *resort to external tools when they encounter problems they cannot solve.* To this end, we need to construct the test set with both positive and negative samples. Positive samples are the queries that can not be solved by LLMs themselves and need tool usage, whereas negative samples are queries that can be directly solved by LLMs and therefore do not necessitate tool usage. For positive samples, we selected a subset of samples from TOOLE and conducted manual validation to confirm whether they would trigger LLMs to use the tool (the process of which is detailed in the Appendix B). As for negative samples, we select three recent instruction datasets, including instructions about downstream tasks (Wang et al., 2022), common-sense questions (Talmor et al., 2019), and high-quality instructions used in LIMA (Zhou et al., 2023b). Similarly, we conducted manual verification to ensure that these requests can be resolved by LLMs' intrinsic capabilities. Specifically, we use the prompt with a query to inquire the LLMs whether need to employ a tool or not, and the output of LLMs should be either "yes" or "no".

### 2.3.2 TOOL SELECTION

**Preliminary.** We propose four subtasks to evaluate LLMs in tool selection *(i.e., *Action* phase (②)). Generally, the prompt comprises a query $q \in Q$ (i.e., the user's input) and a tool list $L_t$ ($L_t \subseteq T$) containing $n$ potential tool candidates. In the single-tool tasks (Sub-task 1∼3), we designate the corresponding tool for query $q$ as $t \in T$. In the multi-tool task (Sub-task 4), this corresponds to $S_t \subset T$ ($|S_t| > 1$). Consequently, we obtain $y_{\text{Action}} \subseteq (L_t \cup \varnothing)$ as the outcome of the tool selection process, where $y_{\text{Action}}$ represents the selected tool(s).

**Sub-task 1: tool selection with similar choices.** The task is designed to challenge LLMs to select the correct tool from a tool list containing similar tools, thereby testing their thorough understanding of tool functionality. Given a query $q$ with its label $t$, we task LLMs with selecting a tool from the specified tool list $L_t$ containing $n$ candidates. To construct $L_t$, we first obtain the embedding of $t$'s description, denoted as $E(t)$, where $E(\cdot)$ represents the embedding function (here, we utilize the text-embedding-ada-002 model (OpenAI, 2023c) to generate embeddings). Denote the most similar tools of $t$ as top-$(n-1)_t$, which are selected based on the cosine similarity of their embeddings: top-$(n-1)_t = \arg \text{top-k}_{t' \in T \setminus \{t\}} \text{sim}(E(t), E(t'))$. Consequently, $L_t = \{t\} \cup \text{top-}(n-1)_t$.

**Sub-task 2: tool selection in specific scenarios.** The objective of this task is to simulate how LLMs perform using tools when they act as controllers of a system (Shen et al., 2023) faced with

---

*We separate the prompt of *Thought* (①) and *Action* (②) to avoid the influence taken from different tool lists.

different scenarios. As LLMs are widely applied across various domains like biomedical domain (**?**) and educational domain (Kasneci et al., 2023), in scenarios where the system caters to diverse demographics or professions (e.g., software engineer (Qian et al., 2023)), its set of tools also varies. This task allows us to explore the performance disparities of LLMs in selecting different kinds of tools, essentially highlighting a form of bias inherent to LLMs (Ferrara, 2023). In such cases, this task examines how effectively LLMs utilize the tools. Given a query $q$ with its label $t$, we specify the tool list $L_t$ containing $n$ candidates according to its corresponding scenarios. This task consists of two types of scenarios: the first one is the popularity-related scenario, and the second one is the group-related scenario. For the popularity-related scenario, we have selected the 5, 10, and 15 most popular tools based on the number of tools it's merged with (refer to Table 9 in the Appendix A.5 for details.) to construct the tool list. As for the group-related scenario, we chose six usual occupations or identities and manually curated a tool list consisting of the 10 most relevant tools for each of them (see Table 10 in the Appendix A.5 for details).

**Sub-task 3: tool selection with possible reliability issues.** The reliability of LLMs' tool selection is of utmost importance. However, issues like hallucination (Ji et al., 2023) and sycophancy (Wei et al., 2023) within LLMs' responses that will negatively impact their selection of tools. Therefore, we introduce sub-task 3. In this task, given a query $q$ and its corresponding tool $t$, we need to construct the tool list $L_t$ and ensure $t \notin L_t$. This aims to assess whether LLMs can answer questions honestly and avoid issues like choosing non-existent tools or selecting unrelated tools. It should be noted that this task closely resembles real-world scenarios, as not all existing tools capable of addressing user queries are present in the tool list controlled by LLMs. To be specific, we obtain the embedding of $t$'s description $E(t)$ and get the *top-$k_t$* similar tools about $t$ as the way in Task 1. Then we randomly sample $n$ tools from the rest tool set $T'$ to construct $L_t$, denoted as $L_t = \{t_1, t_2, ..., t_n\}$ where $t_i \in T'(1 \leq i \leq n)$ and $T' = T\backslash(\{t\} \cup \text{top-}k_t)$. Overall, we remove the ground-truth tool $t$ of query $q$ and $t$'s most similar $k$ tools to keep the tools in $L_t$ not related to $t$ as much as possible.

**Sub-task 4: multi-tool selection.** In addition to testing the selection of single tools, like previous research (Qin et al., 2023b), we set up a task for multi-tool selection which may evaluate the inference ability and more complex semantic comprehension in the tool selection. We tested whether LLMs would correctly choose the specified tools by inputting multi-tool queries. Specifically, given a query $q$ with its related tool set $S_t$ ($|S_t| > 1$), we construct the tool list $L_t$ containing $n$ tool candidates ($n > |S_t|$). Like the candidate selection way in sub-task 3, we obtain each tool $t$'s embedding $E(t)$ where $t \in S_t$, and get the most $k$ similar tools of $t_1, t_2, ..., t_{|S_t|}$, denoted as top-$k_{t_1}$, top-$k_{t_2}$, ... top-$k_{t_{|S_t|}}$. We randomly select $(n - |S_t|)$ tools from $T' = T\backslash(S_t \cup \text{top-}k_{t_1} \cup \text{top-}k_{t_2} \cup ... \cup \text{top-}k_{t_{|S_t|}})$. Finally, these $(n - |S_t|)$ tools and the tools $\in S_t$ consist of the tool list $L_t$. The reason we do not include the most similar tool in $L_t$ like sub-task 3 rather than task 1 is that the multi-tool selection task itself is inherently challenging, and we do not want to further increase the difficulty.

## 3 EXPERIMENTS

### 3.1 EXPERIMENTAL SETUP

**Model selection.** We have chosen eight models that are currently excelling and popular in this field. These models include ChatGPT (OpenAI, 2023a), ChatGLM2 (6B) (THUDM, 2023), Llama2 (7b-chat, 13b-chat) (Touvron et al., 2023), Vicuna (7b, 13b, 33b) (Chiang et al., 2023), Baichuan2 (13b) (Baichuan, 2023) and Koala (13b) (Geng et al., 2023).

**Prompt template and test samples.** Due to the large scale of TOOLE, we sample from it as our test set (more details are shown in Appendix C). For a better understanding of the importance of tool usage and to tell LLMs when need to use tools, we add the reasons for tool usage in the prompt template of Thought (①) part. We show the detailed prompt template in Appendix D.2. We also conducted few-shot learning experiments for the first three tasks and details of the experimental design can be found in Appendix C.4.

**Metrics.** For the awareness of tool usage evaluation, we use accuracy, recall, precision, and F1 score as the metrics. For tool selection, we propose the Correct Selection Rate (CSR) to calculate the percentage of correct selection action. Denote the output results for all queries as $Y = \{y_1, y_2, ...\}$, for a specific output $y$, we use $A(y)$ to denote the tool(s) that the model chooses from the tool list. The CSR is computed as follows:

Table 2: The results for the awareness of tool usage test. We use accuracy (Acc.), precision (Pre.), recall (Rec.), and F1 score (F1) as evaluation metrics. And F1 $\Delta$ is the percentage change of F1 Score between zore-shot and five-shot, as calculated by $F1_{x=5} - F1_{x=0}$

| Metrics | | ChatGPT | ChatGLM2 | Llama2-7b | Llama2-13b | Vicuna-7b | Vicuna-13b | Vicuna-33b | Koala-13b |
|---|---|---|---|---|---|---|---|---|---|
| | **Acc.** | 74.85 | 54.56 | 53.88 | 51.94 | 52.43 | 61.17 | 65.05 | 54.95 |
| $x=0$ | **Pre.** | 69.63 | 61.90 | 52.71 | 80.00 | 74.51 | 82.49 | 79.92 | 55.53 |
| | **Rec.** | 88.16 | 40.39 | 84.85 | 6.21 | 7.38 | 28.35 | 40.19 | 52.62 |
| | **F1** | 77.81 | 48.88 | 65.03 | 11.53 | 13.43 | 42.2 | 53.49 | 54.04 |
| | **Acc.** | 79.71 | 56.02 | 50.39 | 57.86 | 56.12 | 55.15 | 61.55 | 54.66 |
| $x=5$ | **Pre.** | 73.98 | 64.32 | 50.55 | 63.31 | 56.49 | 52.92 | 57.85 | 52.61 |
| | **Rec.** | 91.65 | 51.46 | 99.03 | 47.57 | 54.95 | 96.89 | 85.83 | 97.86 |
| | **F1** | 81.87 | 57.17 | 66.93 | 54.32 | 55.71 | 68.45 | 69.12 | 68.43 |
| **F1 $\Delta$** | | 4.06 ↑ | 8.92 ↑ | 1.90 ↑ | 42.79 ↑ | 42.28 ↑ | 26.25 ↑ | 15.63 ↑ | 14.39 ↑ |

$$\text{CSR} = \frac{1}{|Y|} \sum_{y \in Y} \mathbb{I}\left( A(y) = \begin{cases} t & \text{for Task 1,2} \\ \varnothing & \text{for Task 3} \\ S_t & \text{for Task 4} \end{cases} \right) \tag{1}$$

## 3.2 Results Analysis

Through the experiment results, we have gained the following conclusions:

**Even under the few-shot prompts, the majority of LLMs still perform poorly in tool usage awareness.** In Table 2, we observe that under the zero-shot prompt, only ChatGPT has both accuracy and F1 score exceeding 70%, while the performance of other models is relatively poor, with the F1 score of llama2-13b being only 11.53%. Under the five-shot prompt, some models show significant improvement in F1 scores, for example, llama2-13b increased by 42.79%, and vicuna-7b by 42.28%. This indicates that though few-shot learning generally improves the performance of LLMs in tool usage awareness, they still lack sufficient tool usage awareness.

**When selecting similar tools, there is a significant performance disparity among existing LLMs, and the improvement brought by few-shot prompts is limited.** Table 3 shows that under the zero-shot prompts, the best-performing LLM is Vicuna-7b, with nearly a 30% difference compared to the worst-performing Llama2-13b. The gap between the best-performing ChatGPT and the worst-performing Llama2-13b still exceeds 20% under 5-shot prompts. Additionally, the maximum improvement brought by 5-shot prompts does not exceed 7%. Moreover, the performance of Vicuna-7b even declined by 10% under the five-shot condition, suggesting a potential bias in its 0-shot performance, which reflects either a lack of robustness or over-sensitivity of the model.

**LLMs still face serious challenges in dealing with reliability issues, for instance, reducing hallucination.** As seen from Table 3, although few-shot prompts improve the performance of all LLMs, the CSR of most LLMs remains below 20%. We find that LLMs sometimes fabricate non-existent tools, a severe hallucination issue that has a significant impact on LLM-based agents. Additionally, the potential sycophancy of LLMs may lead them to avoid returning a "none" answer, instead choosing irrelevant tools to respond to users.

Table 3: The CSR (%) for tool selection with similar choices and with possible reliability issues. $\Delta$ is the percentage change of CSR between zore-shot and five-shot, as calculated by $\text{CSR}_{x=5} - \text{CSR}_{x=0}$.

| Metric | | ChatGLM2 | ChatGPT | Llama2-7b | Llama2-13b | Vicuna-7b | Vicuna-13b | Vicuna-33b | Koala-13b |
|---|---|---|---|---|---|---|---|---|---|
| | $x=0$ | 54.17 | 69.05 | 45.95 | 44.06 | **73.46** | 58.23 | 53.96 | 56.34 |
| **Similar** | $x=5$ | 57.44 | **72.94** | 51.12 | 49.85 | 63.67 | 63.15 | 60.54 | 60.85 |
| | $\Delta$ | 3.27 ↑ | 3.89 ↑ | 5.17 ↑ | 5.79 ↑ | -9.79 ↓ | 4.92 ↑ | 6.58 ↑ | 4.51 ↑ |
| | $x=0$ | 6.63 | **50.35** | 0.90 | 2.31 | 1.50 | 2.51 | 2.81 | 1.70 |
| **Reliability** | $x=5$ | 15.68 | **78.49** | 2.51 | 5.93 | 1.81 | 3.42 | 3.11 | 5.83 |
| | $\Delta$ | 9.05 ↑ | 28.14 ↑ | 1.61 ↑ | 3.62 ↑ | 0.31 ↑ | 0.91 ↑ | 0.30 ↑ | 4.13 ↑ |

**LLMs perform poorly in processing long texts.** From Figure 4, we can see that the CSR of almost all LLMs decreases as the length of the tool list increases, especially in the range from top 5 to top 10. This indicates that LLMs still need improvement in understanding long texts. **LLMs exhibit imbalances and biases in tool selection across different scenarios.** For example, in Figure 5, LLMs generally have a higher CSR in tool selections related to the elderly and artists & designers, while their CSR is lowest for tools related to students. This means that developers still need to enhance the generalization capabilities of LLMs. At the same time, for downstream applications, it is best to choose suitable LLMs based on different applied fields.

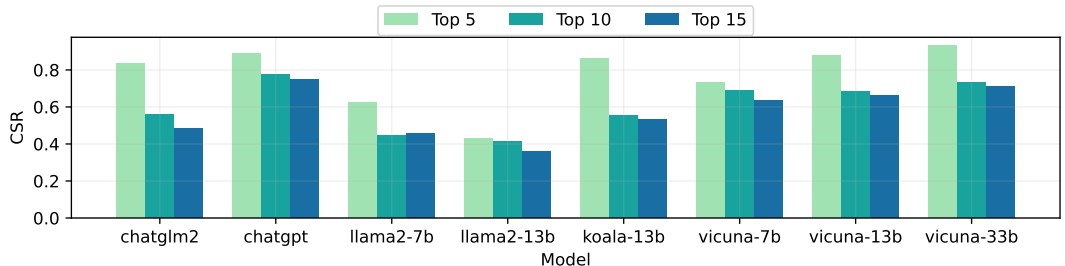

Figure 4: The CSR results (%) of top $n$ ($n$=5,10,15) tool in different scenarios.

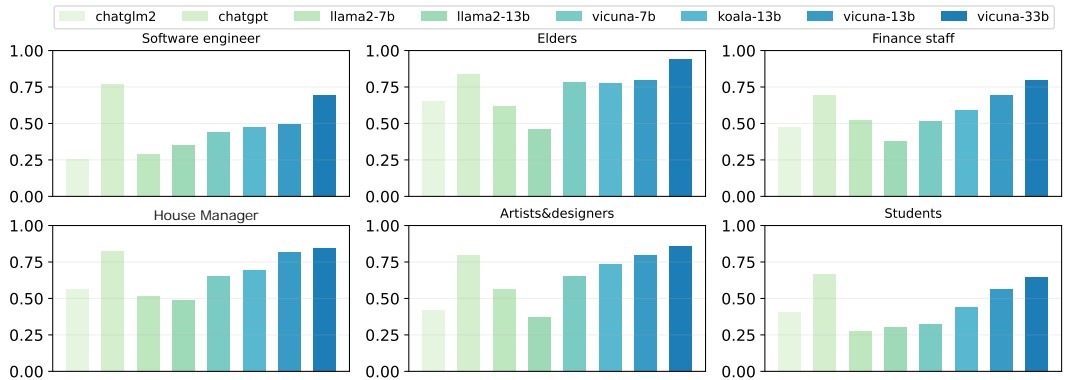

Figure 5: The CSR result (%) of tool selection in specific scenarios.

Table 4: Multi-tool selection results. We evaluate LLMs' performance based on two kinds of prompt templates: one is telling LLMs to choose zero, one, or two tools (i.e., multi-choice), while another is forcing LLMs to choose two tools (i.e., one-choice). We consider the different kinds of CSR (%) for the former one: the LLM selects two correct tools (2/2 CSR), selects only one tool and it's correct (1/1 CSR), and selects two but only one is correct (1/2 CSR).

| | Metric | ChatGPT | ChatGLM2 | Llama2-7b | Llama2-13b | Vicuna-7b | Vicuna-13b | Vicuna-33b | Koala-13b |
|---|---|---|---|---|---|---|---|---|---|
| | **2/2 (CSR)** | 88.28 | 20.20 | 35.69 | 81.49 | 44.06 | 83.70 | 48.69 | 39.03 |
| **Multi-choices** | **1/1 (CSR)** | 3.03 | 36.57 | 21.98 | 0.00 | 25.55 | 1.01 | 48.49 | 1.61 |
| | **1/2 (CSR)** | 1.01 | 13.94 | 8.87 | 11.07 | 7.04 | 8.05 | 0.20 | 20.32 |
| **One-choice** | **CSR** | 88.53 | 23.34 | 57.34 | 77.87 | 64.34 | 78.47 | 91.15 | 25.10 |

**There are significant performance differences among LLMs in multi-tool selection.** As shown in Table 4, ChatGPT, the top-performing model, outperforms ChatGLM2, the worst-performing model, by nearly 70%, highlighting the variability in the capabilities of different language models for this task. Furthermore, the most common error made by the models is omitting tool selection, such as in the case of Vicuna-33b, which only selected one tool in 48.49% of cases. Moreover, **several LLMs overly rely on the explicitly specified number of tools they should select in the prompts.** As shown in Table 4, when explicitly instructed to return two tools, Vicuna-33b's correct selection rate increased to over 90%, and Vicuna-7b also improved by over 20%. This indicates that these

Table 5: Error analysis results. The Top@$k$ metric quantifies the proportion of incorrect choices by the model that are ranked within the Top@$k$ positions of the similarity-ranked list.

| Top@k | ChatGPT | ChatGLM2 | Llama2-7b | Llama2-13b | Vicuna-7b | Vicuna-13b | Vicuna-33b | Koala-13b |
|---|---|---|---|---|---|---|---|---|
| **Top@1** | 18.44 | 19.89 | 14.37 | 15.12 | 15.03 | 15.91 | 16.62 | 15.04 |
| **Top@3** | 34.29 | 36.48 | 34.08 | 34.94 | 34.43 | 35.94 | 35.09 | 35.34 |
| **Top@5** | 47.26 | 43.63 | 51.41 | 49.81 | 50.55 | 48.15 | 48.28 | 49.62 |

LLMs still possess good multi-tool selection capabilities but require prior knowledge, which makes it challenging to apply in LLM-based agents.

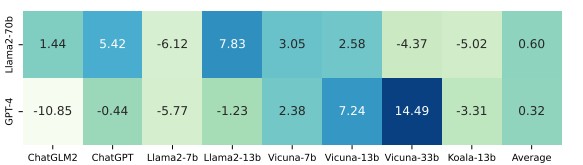

Figure 6: Percentage change on the description rewritten by Llama2-70b and GPT-4.

**Error analysis.** We further investigate the reasons for errors in LLM's tool selection. We employ the Top@$k$ metric to analyze failure cases in tool selection with similar choices, as shown in Table 5. It suggests that, despite being incorrect, the choices made by the model often retain a degree of similarity to the correct tool. In general, all LLMs have a nearly 50% chance of choosing a tool from the Top@5 most similar to the correct tool, and more than a 15% chance of choosing the most similar one (i.e., Top@1). This indicates that there is still significant room for improvement in tool selection with LLMs.

**Insights for Tool Developer.** We also investigated the relationship between tool descriptions and CSR. We calculated CSR for the queries corresponding to $t$ and visualized them in Figure 7. There are two categories of tools: those that have been decomposed and merged (i.e., new tools) and those that have not been merged or decomposed (i.e., original tools). From the figure, we can draw the conclusion: **The more detailed the description, the more efficient tool selection.** As shown by the fitted line, as the length of the description increases, the CSR continuously increases, indicating that detailed descriptions can help LLMs better understand the functionality of tools, thus improving the accuracy of tool selection. Additionally, as shown in Figure 6, we built upon the original description by having two proficient LLMs rewrite it and then observed the performance changes of eight LLMs on the new descriptions. Different rewritten LLMs yielded varying benefits for different groups. For instance, descriptions rewritten by Llama2-70b resulted in a 7.83% improvement for llama2-13b, but did not significantly enhance the performance of the Vicuna series models. In contrast, descriptions

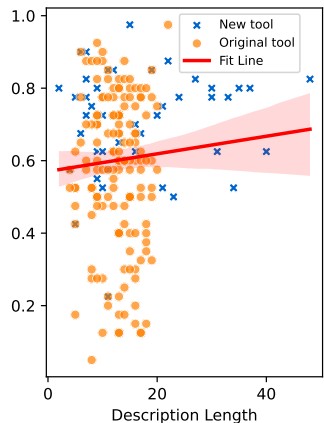

Figure 7: The CSR of tool selection and description length.

rewritten by GPT-4 caused a sharp decline in the performance of ChatGLM and Llama2 series, while significantly boosting the Vicuna series, possibly due to the Vicuna series' training corpus being largely sourced from ShareGPT (ShareGPT, 2023). **Therefore, we strongly recommend that tool developers choose an appropriate rewrite model for generating new descriptions based on the downstream LLM the tool will apply to.**

## 4 CONCLUSION

In this paper, we introduce METATOOL, a benchmark for evaluating LLMs based on their tool usage awareness and tool selection capabilities. We propose TOOLE within the benchmark, which contains diverse queries to trigger LLMs to use tools. We found that most LLMs lack good tool usage awareness and exhibit a significant gap from real intelligent agents in tool selection.

ACKNOWLEDGEMENT

Lichao Sun and Yue Huang are supported by the National Science Foundation Grants CRII-2246067 and Microsoft Accelerate Foundation Models Research Award.

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

# Appendix

## Table of Contents

## A   TOOLE DATASET DETAILS

In this section, we show the details of TOOLE, including how we solve the overlapped issue (Section A.1), guidelines for human validation (Section A.3), and the statistics of TOOLE (Section A.5).

### A.1   DETAILS OF OVERLAPPED ISSUE

**Operation pipeline.**

(1) *Embeddings and Hierarchical Clustering.* We initiated by generating embeddings for tool descriptions using the text-embedding-ada-002 model (OpenAI, 2023c), an API provided by OpenAI, aiming to perform hierarchical clustering (Sneath et al., 1973) on different tools, based on the similarity of their embeddings, to reveal underlying patterns among them. (2) *Tool Merging and Decomposition.* Based on the results of clustering, we manually merged and decomposed the data. Specifically, several popular topics (e.g., news, weather) were identified based on their functions if they overlapped functionality with other tools (an example is shown in Figure 8). The criteria for merging and decomposition revolved around whether such tools are commonly encountered and make practical sense in daily life. For instance, it is logical to merge a tool that offers both flight and train ticket bookings with another tool that solely focuses on hotel

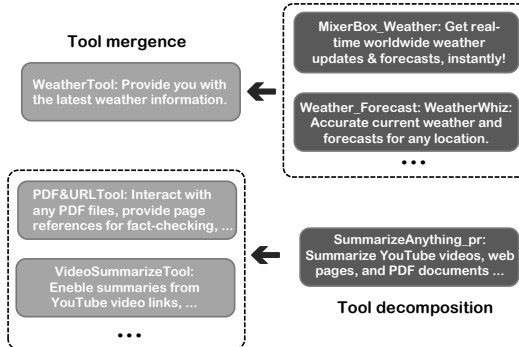

Figure 8: Two examples of tool mergence and decomposition.

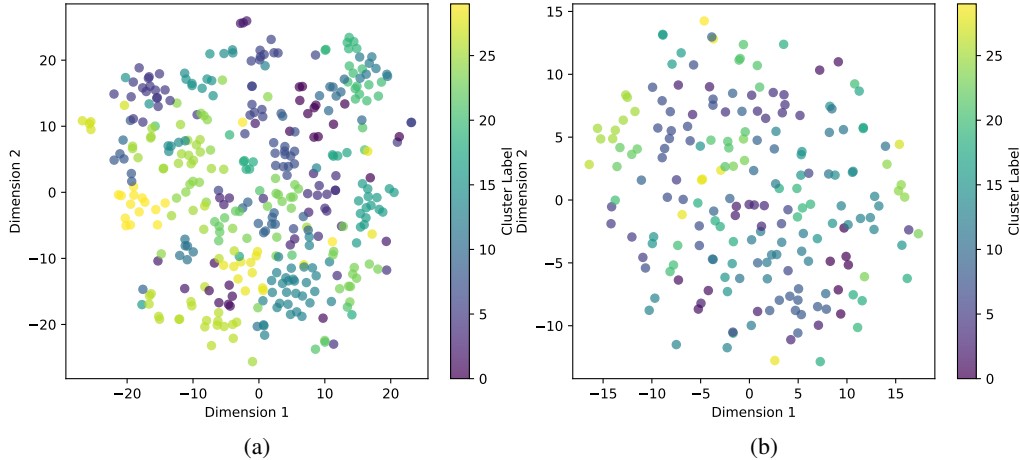

Figure 9: t-SNE (Van der Maaten & Hinton, 2008) visualization of original tool description embedding (a) and new tool description embedding (b).

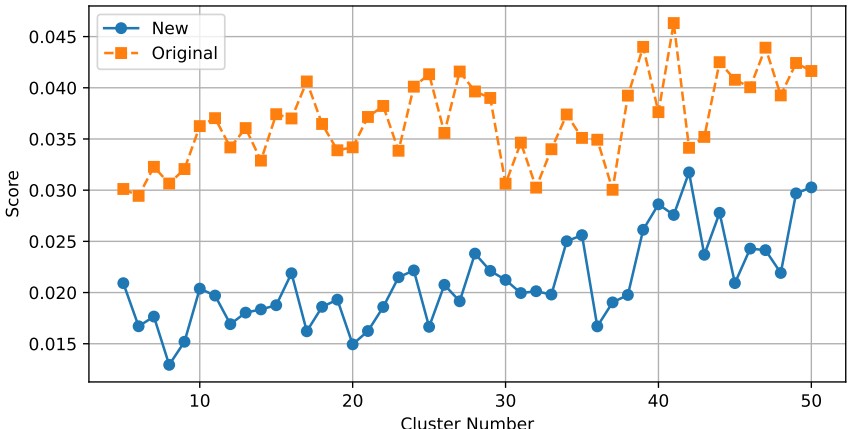

Figure 10: Silhouette score (Rousseeuw, 1987) of new tool description embedding and original tool description embedding in different cluster numbers.

reservations. For merged tools, we only needed to modify the original labels; for decomposed tools, we manually assigned original queries to the appropriate decomposed tools and changed their labels accordingly. We manually created the names of new tools and employed ChatGPT to generate descriptions for both merged and decomposed tools, based on the initial descriptions of these tools. (3) *Similarity Verification and Human Validation.* We iterated each tool and searched for the ten most similar tools by its description embedding, then we checked whether the tool could be further merged or decomposed with the tools in $L_{sim}$.

## A.2 EFFICIENCY OF THE OPERATION

To evaluate the effectiveness of our operations in solving overlapped issues, we use the silhouette coefficient (Rousseeuw, 1987) to measure the degree of functional overlap between tools. Based on it, we compare the changes in the silhouette coefficients before and after the operations. Specifically, we aim to significantly reduce the functional overlap between tools after merging and decomposition to achieve a more uniform distribution of tool functionalities in the embedding space. We embed the tool description before (390 tools) and after the operations (195 tools) and compared the changes in the silhouette coefficients under the same number of clusters. A greater silhouette coefficient indicates better clustering performance and higher functional overlap between tools, while a smaller coefficient suggests lower overlap, reflecting the effectiveness of the operations.

In Figure 10, we present the variation of silhouette coefficients with changes in the number of clusters. It can be observed that the silhouette coefficients of tools after the operations are significantly smaller than before, indicating that the operations have made the distribution of tools more uniform and effectively reduced tool functionality overlap. Additionally, in Figure 9, we visualize the results using t-SNE (Van der Maaten & Hinton, 2008) when the number of clusters is 30. It is evident that the distribution on the left side is more uniform compared to the right side.

### A.3    GUIDELINES FOR HUMAN VALIDATION

We conducted rigorous manual evaluations to ensure the integrity and quality of TOOLE. We established the following rules to guide the manual evaluation:

- Low-quality tool descriptions. In some cases, ChatGPT was unable to understand the purpose of a tool due to low-quality or overly brief tool descriptions. We conducted a manual review of these descriptions and eliminated tools with unclear or low-quality explanations.

- High repetition queries. Since we generated multiple queries for a single tool in one batch, some batches had issues with high query repetition. To address this problem, we selected one query and removed the others.

- Queries contain tool name. The inclusion of a tool's name in a query can significantly bias our evaluation as an obvious hint for all tasks in METATOOL. Therefore, we removed queries containing the tool's name. For example, 'How can I calculate my MBTI type through [tool name]?'

- Calculation-related tools. TOOLE contained numerous tools related to calculations. For simple calculations (e.g., 'What is the value of sin 30 degrees?' or '7 * 9 = ?'), LLMs can perform them without the need for a tool. However, for complex calculations, recent research (Yuan et al., 2023a) suggests that LLMs still perform poorly. For queries corresponding to calculation-related tools, we removed queries involving simple calculations and retained those involving complex calculations.

- Tool retrieval-related tools. We found that some tools were designed for users to retrieve other tools. This kind of tool conflicted with our task, so we removed these tools.

- AI comprehensive tools. We identified some AI comprehensive tools that encompassed a wide range of AI-related tools, making them impractical for our evaluation. Therefore, we removed these tools.

- Mentions of 'ChatGPT' in queries. Some queries included the term 'ChatGPT,' for example, 'Hi, ChatGPT! ...' We uniformly replaced 'ChatGPT' with 'Chatbot'."

### A.4    FLEXIBILITY OF TOOLE

Sometimes, when an LLM undergoes specialized training in a particular domain (Wu et al., 2023b; Wang et al., 2023a; Qi et al., 2023; Wang et al., 2023b; Yang et al., 2023a), its capabilities improve significantly, and in some cases, some LLMs are also capable of handling various types of information (e.g., images or audio) (Liu et al., 2023; Chen et al., 2023; Wang et al., 2023d; Lyu et al., 2023; Li et al., 2023b). These improvements render some external tools that may not be necessary for some LLMs in the future. Therefore, we have annotated the reasons why LLMs need to use these tools to solve user problems.

Specifically, we use four kinds of motivation for tool usage (some examples are shown in Table 6): A. Solving issues with real-time or external data, databases, or APIs. B. Handling specialized inputs/outputs. C. Enhancing domain tasks beyond LLM's capabilities. D. User

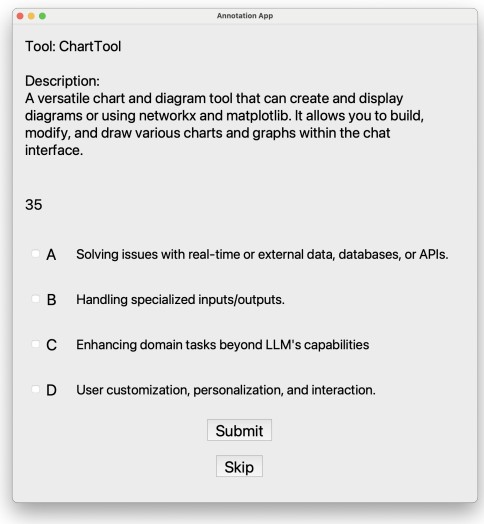

Figure 11: The motivation labeling interface.

Table 6: Possible reasons for the motivation of tool usage.

| Tool | Description | Reason |
|---|---|---|
| airqualityforeast | Planning something outdoors? Get the 2-day air quality forecast for your city. | A |
| Now | Get Google Trends. In Japan, you can also get Twitter trends and search Twitter keywords. | A |
| abc_to_audio | Converts ABC music notation to WAV, MIDI, and PostScript files. | B |
| ChatOCR | The best way to read text from from any document. ChatOCR will scan and read aloud any text document you provide. | B |
| FinanceTool | Begin an exciting journey through time, interact with historical events, and learn about the past in a fun and engaging way. | C |
| LawTool | Enables quick search functionality for relevant laws. | C |
| TicTacToe | Playing a game of Tic Tac Toe with varying board sizes. You can submit your move and get the AI's response move. | D |
| Planfit | Get your tailored workout plan and instructions with videos - AI-powered Workout Coach, Planfit. | D |

customization, personalization, and interaction.

We have enlisted the expertise of two experts to annotate the usage motivations of tools in TOOLE (the annotation interface is shown in Figure 11).

### A.5 OTHERS STATISTICS OF TOOLE

**Data quantity.** Due to the constraints of API usage costs, we exclusively employ GPT-4 for direct diverse generation, utilizing ChatGPT for all other forms of generation techniques. This strategy yielded a total of 29,000 dataset entries. Following a meticulous human review process, we ultimately curated the TOOLE dataset, culminating in a collection comprising 20,881 entries. A comprehensive overview of dataset statistics can be found in Table 11.

**Length distribution.** Figure 12 displays the distribution of dataset lengths. It can be observed that the majority of the data falls within 40 words or less, aligning with the typical question lengths in people's daily lives.

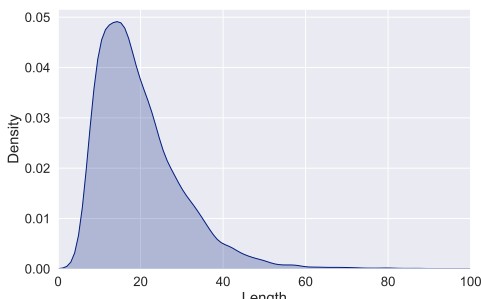

Figure 12: Density distribution of all queries' length.

**Visualization and release.** We used Nomic AI (AI, 2023) to embed user queries in the TOOLE dataset, then clustered the embeddings, and finally visualized the results. The visualization is shown in Figure 13, and you can view it through the following link: https://atlas.nomic.ai/map/a43a6a84-4453-428a-8738-2534d7bf0b89/b2b8134b-a37e-45d2-a0d9-765911f27df6.

**Dataset comparison.** As shown in Table 7, we compare the other datasets with TOOLE. Compared to other datasets, we believe that ToolE has two main advantages: (1) Our dataset exhibits greater diversity, and this diversity is tailored to real user scenarios, such as variations in expression style, mood, and level of detail. We employ various prompt methods to induce LLMs to generate a more diverse range of user inputs, ensuring that TOOLE covers a broad spectrum of inputs resembling those of actual users. (2) By employing a pipeline process to address overlapped issues, we can ensure the rigor of the data. As outlined in Appendix A.1, we employ multiple steps to address overlapped issues, ensuring that there is no overlap between tools, which is crucial for maintaining the quality of the dataset.

**Tool description.** These original tool descriptions encompass two distinct categories: machine-readable descriptions and user-facing descriptions. The machine-readable descriptions prioritize considerations such as token context length or keyword incorporation, aiming to enhance tool prompting within an 8,000-character limit. Conversely, the user-facing descriptions offer succinct

Table 7: Comparison of previous work and METATOOL.

| Dimension | APIBank Li et al. (2023d) | ToolLLM Qin et al. (2023b) | ToolAlpaca Tang et al. (2023) | GPT4Tool Yang et al. (2023c) | ToolQA Zhuang et al. (2023) | ToolE (Ours) |
|---|---|---|---|---|---|---|
| **Multi-Tool** | ✗ | ✔ | ✗ | ✗ | ✗ | ✔ |
| **Real Scenario** | ✔ | ✔ | ✗ | ✔ | ✔ | ✔ |
| **Overlapped Issue Solved** | ✗ | ✗ | ✗ | ✗ | ✗ | ✔ |
| **Diversity Generation** | ✗ | ✔ | ✗ | ✔ | ✔ | ✔ |

Table 8: Data examples in TOOLE.

| Example | Type |
|---|---|
| I can't seem to remember anything I study. I need help with learning and retaining information effectively. | Emotion-angry |
| I'm feeling really down today, can you summarize this YouTube video for me? [Link to YouTube video] | Emotion-depressed |
| Good day! I'm hoping to find some amazing bargains and discounts today. Can you guide me in the right direction? | Emotion-happy |
| Hey Chatbot, I'm looking for a new pair of sneakers on GoFynd. Can you help me find the latest designs and recommend some popular brands? | Emotion-excited |
| I'm interested in Japanese cosmetics. Can you recommend some good brands? | Direct-request |
| Draw a state diagram for a vending machine. | Direct-order |
| Twitter Trends: What are the trending topics on Twitter in Japan? | Keyword |
| Could you recommend some popular shopping malls or markets in Singapore that offer a wide range of products, including local and international brands, diverse food options, and unique shopping experiences? | Details |

Table 9: Top 15 tools ranked by the number of merged tools.

| Tool Name | Merged Tools | Tool Name | Merged Tools | Tool Name | Merged Tools |
|---|---|---|---|---|---|
| FinanceTool | 22 | ResearchFinder | 7 | TripAdviceTool | 6 |
| ProductSearch | 19 | NewsTool | 7 | WeatherTool | 6 |
| JobTool | 12 | RepoTool | 6 | HousePurchasingTool | 5 |
| TripTool | 10 | ResearchHelper | 6 | Discount | 5 |
| PDF&URLTool | 8 | CourseTool | 6 | MusicTool | 5 |

Table 10: The tool lists of different scenarios.

| Scenario | Tools | Ref. |
|---|---|---|
| Software Engineer | RepoTool, AI2sql, SSH, AutoInfra1, noteable, dart, hackit_web_scanner, LarkBaseImporter, webhooks, universal | (von Werra & Allal, 2023; Odendahl, 2023; Welsh, 2023; Bridgwater, 2023; Mendelevitch, 2023) |
| Elders | NewsTool, PolishTool, CharityTool, MapTool, MemoryTool, WeatherTool, RestaurantBookingTool, DietTool, NotesTool, TripAdviceTool | (Science & Council, 2019; Mears, 2015) |
| Finance Staff | FinanceTool, ChartTool, PDF&URLTool, NotesTool, ExchangeTool, CreditYelp, LawTool, DataRetrievalTool, fundsdbsearch, CompanyInfoTool | (Raieli, 2023; Zhang, 2023; Vidal, 2023; Yang et al., 2023b; Team, 2023) |
| House manager | tira, Discount, ProductSearch, ABCmouse, RestaurantBookingTool, IndoorPlants, recipe_retrieval, HouseRentingTool, TripTool, CreditYelp | (Empresaria, 2023) |
| Students | CourseTool, ResearchFinder, ResearchHelper, speak, noteable, search, MemoryTool, NotesTool, MixerBox_Translate_AI_language_tutor, ABCmouse | (Xiao, 2023; Awan, 2023; Hosni, 2023; Writings, 2023) |
| Artists&designers | placid, find_agency, ArtCollection, ChartTool, storybird_stories, MediaModifyTool, PolishTool, MusicTool, ImageSearch, BookTool | (Kashyap, 2023; BIRCH & VALEANU, 2023) |

Table 11: Dataset Statistics of TOOLE.

| Generation method | Model | Sample number |
|---|---|---|
| Direct generation | ChatGPT, GPT-4 | 11,700 |
| Emotional generation | ChatGPT | 7,800 |
| Keyword generation | ChatGPT | 1,950 |
| Details generation | ChatGPT | 7,800 |
| Multi-tool generation | ChatGPT, GPT-4 | 1624 |
| After checking → 21,127 (20630 single-tool + 497 multi-tool) | | |

and simplified explanations of each tool's functionality. While the majority of tools share identical descriptions across both categories, we opt to adopt the user-facing descriptions as the definitive tool descriptions. This choice is informed by the tendency of machine-readable descriptions to be overly verbose, often delving into instructing the language models on how to handle tool input and output – aspects that are not pertinent to our benchmark.

**Generation times.** For each type of original tool, we perform two rounds of direct diverse generation, producing ten queries each time. In the case of emotional generation, we generate five samples for each of the four distinct emotions. For keyword generation, we extract five keywords from the tool's description and subsequently formulate a query for each identified keyword. Concerning details generation, the number of samples generated aligns with that of the direct diverse generation.

## B  TEST SET FOR EVALUATION ON THE AWARENESS OF TOOL USAGE

To assess to what extent LLMs are aware of their limitations, we construct the dataset for the awareness of tool usage by merging the positive samples from a subset of TOOLE dataset and the negative samples from some subsets of public datasets. To exclude the ambiguous situation where the queries in TOOLE can be solved either with or without the assistance of external tools, we manually check the output of the queries by feeding them directly into the LLMs and retain those whose responses are unsatisfactory(i.e. the model apologizes, the response contains errors, and etc). In this way, we verify that solving the queries in our selected subset(containing 515 samples) is beyond

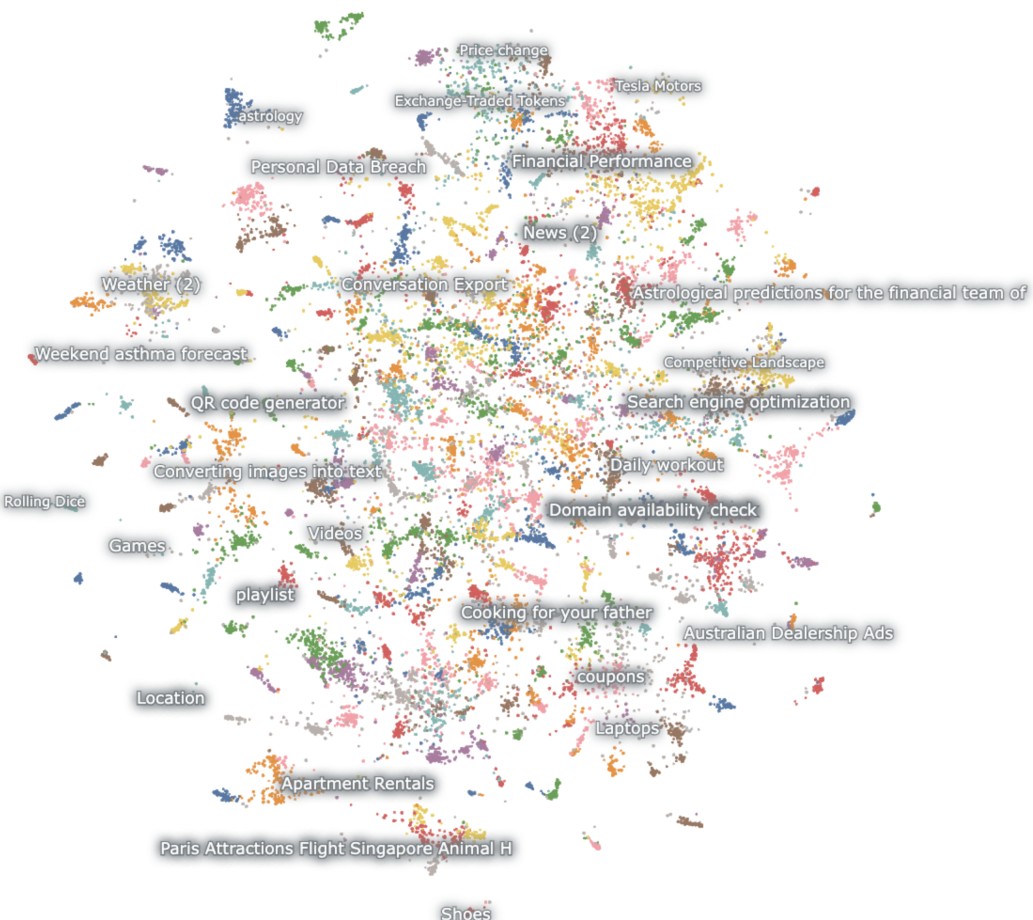

Figure 13: TOOLE embedding visualization.

the capacity of the existing LLMs and the queries can therefore be treated as positive samples when evaluating the awareness of tool usage. Meanwhile, we collect negative samples of equal size from the public datasets of daily conversations (Wang et al., 2022; Zhou et al., 2023b) and common sense (Talmor et al., 2019), the queries of which are expected to be answered by LLMs without the help of external tools.

In order to gain a clearer understanding of the Tool Usage Awareness dataset, we will describe how we selected positive and negative samples for the dataset. Firstly, we categorized user queries into three types:

- The first type is "Queries must be solved by tools" (positive), such as multimodal input and real-time information retrieval.
- The second type is "Queries can be solved well by all LLMs" (negative), such as telling jokes, basic conversational functions, sentiment classification, and other basic NLP tasks.
- The third type represents the middle ground between the first and second types of user queries, i.e., queries we hope LLMs can solve but currently cannot, such as complex calculations, long text summarization, and information extraction.

With the aforementioned types of user queries, we classified user queries into positive or negative samples using human evaluation and model checking.

(1) Human evaluation: For the first and second types, we determined the samples through unanimous agreement from two human experts and referenced the four reasons in Appendix A.4 for selection.

(2) Model Checking: Regarding the third type of user queries, our objective is to find those within this category that can be solved well by all LLMs (classified as negative) and those that none of the LLMs can solve (classified as positive). We discarded user queries that only a portion of the LLMs can solve them. We conduct validation in the following two steps:

We initially input the queries into GPT-4. Since GPT-4 currently has the best performance in terms of utility, *if GPT-4 declines to answer (i.e., unable to solve the problem or refuse to answer), we classify it as a positive query.*

Following the above operation, if the query is not classified as positive, we conducted inference on eight LLMs and then evaluated the answers through two human experts. *If all output from LLMs solves the problem well, we classify them as negative queries.*

## C EXPERIMENTAL SETTINGS

### C.1 MODELS AND TEST SAMPLES

We conducted extensive experiments on eight commonly used LLMs including ChatGPT (OpenAI, 2023a), ChatGLM2 (6B) (THUDM, 2023), Llama2 (7b, 13b) (Touvron et al., 2023), Vicuna (7b, 13b, 33b) (Chiang et al., 2023), Baichuan2 (13b) (Baichuan, 2023) and Koala (13b) (Geng et al., 2023). The temperature parameter for these models was uniformly set to 0.0 in our experiments. For tool selection with similar choices and with possible reliability issues, we sampled five samples for each tool, resulting in a total of 975 samples. For tool selection in specific scenarios, we sampled 20 samples for each tool within each scenario. This means that for popularity-related scenarios, namely the top 5, 10, and 15 scenarios, we obtained 100, 200, and 300 samples, respectively. As for group-related scenarios, each scenario yielded 200 samples. In the case of"multi-tool selection, given the relatively small dataset size, we utilized the entire set of samples for testing.

### C.2 ANSWER MATCHING

Due to the model's practice of providing an explanatory context for its responses rather than directly outputting answers, it is necessary to perform answer matching (obtaining "yes," "no," or the name of a tool) for a more accurate evaluation.

**Tool usage awareness.** We use the following rules to match the results:

- If "yes" is present in the sentence and "no," "not," or "don't" is absent from the sentence: The answer is "yes."
- If "no" is present in the sentence and "yes" is absent from the sentence: The answer is "yes." If phrases like "not seem necessary," "not think it is necessary," "not need to use," "not necessary to use," and "do not think I need to use" are present in the sentence: The answer is "no."
- If phrases like "I need to use," "I think it is necessary," "I may need to use," "I would need to use," "I believe it is necessary to use," "would need access," "might be necessary to use," "I might need to use," "tools would be necessary," "may be necessary to use," "be beneficial to use," "I will need to use," "might need to use," "would need to rely," and "may need to access" are present in the sentence: The answer is "yes."

For cases not covered by the above rules, the answers are analyzed manually.

**Tool selection.** We match the names of tools from the results that contain a tool list. For single-tool tasks, if no matches are found, it is recorded as "None." If one match is found, and "None" is not present in the results, it is compared with the ground-truth label. If more than two matches are found, manual analysis is performed. For multi-tool tasks, if the number of matches is less than two, it is considered an incorrect answer. If the number of matches is equal to two, it is compared with the ground-truth label. If more than two matches are found, the answers are manually evaluated.

### C.3 TASK COMPARISON

In order to have a more intuitive understanding of the different subtasks in tool selection, we show the comparison of the four sub-tasks in Table 12.

Table 12: Comparison of four tasks in tool selection.

| Task | Tool list | Ideal output |
|---|---|---|
| Tool selection with similar choices | Tool $t$ and its most similar $(n-1)$ tools. | $t$ |
| Tool selection in specific scenarios | Specified tools in a certain scenario. | $t$ |
| Tool selection with possible reliability issues | The tools randomly chosen are from the remaining set of tools, which excludes tool $t$ and its $k$ most similar tools. | $\varnothing$ |
| Multi-tool Selection | The tools randomly chosen are from the remaining set of tools, which excludes the union set of $k$ most similar tools of $t \in S_t$, where $S_t$ is the ground-truth tool set. | $S_t$ |

Table 13: CSR (%) results of zero-shot and five-shot in different scenarios. $\Delta$ is the percentage change of CSR between zore-shot and five-shot, as calculated by $\text{CSR}_{x=5} - \text{CSR}_{x=0}$. Abbreviation: Finance Staff (Finan.), House manager (Home.), Software Engineer (Soft.), Student (Stud.), and Artist & Designer (Artis.).

| Scenario | | Finan. | Home. | Soft. | Stud. | Artis. | Elders | Top 5 | Top 10 | Top 15 |
|---|---|---|---|---|---|---|---|---|---|---|
| **ChatGLM2** | $x=0$ | 46.70 | 57.59 | 28.50 | 40.31 | 42.78 | 68.42 | 82.29 | 56.19 | 48.07 |
| | $x=5$ | 62.57 | 62.94 | 41.53 | 34.97 | 45.64 | 69.66 | 86.17 | 57.22 | 50.54 |
| | $\Delta$ | 15.87 | 5.35 | 13.03 | -5.34 | 2.86 | 1.24 | 3.88 | 1.03 | 2.47 |
| **ChatGPT** | $x=0$ | 69.70 | 82.23 | 76.70 | 66.67 | 80.00 | 83.76 | 88.89 | 77.39 | 75.08 |
| | $x=5$ | 74.24 | 86.73 | 69.11 | 72.45 | 87.00 | 89.45 | 91.00 | 80.50 | 77.33 |
| | $\Delta$ | 4.54 | 4.50 | -7.59 | 5.78 | 7.00 | 5.69 | 2.11 | 3.11 | 2.25 |
| **Koala-13b** | $x=0$ | 59.32 | 69.47 | 47.69 | 43.75 | 73.58 | 77.78 | 86.42 | 55.56 | 52.98 |
| | $x=5$ | 51.83 | 61.85 | 46.99 | 53.44 | 64.38 | 82.35 | 81.71 | 69.82 | 52.59 |
| | $\Delta$ | -7.49 | -7.62 | -0.70 | 9.69 | -9.20 | 4.57 | -4.71 | 14.26 | -0.39 |
| **Llama2-7b** | $x=0$ | 52.04 | 57.07 | 30.00 | 32.99 | 59.09 | 57.14 | 65.66 | 45.73 | 44.86 |
| | $x=5$ | 64.97 | 62.12 | 42.35 | 40.00 | 69.00 | 76.17 | 67.68 | 47.24 | 54.08 |
| | $\Delta$ | 12.93 | 5.05 | 12.35 | 7.01 | 9.91 | 19.03 | 2.02 | 1.51 | 9.22 |
| **Llama2-13b** | $x=0$ | 35.18 | 47.00 | 34.85 | 35.00 | 30.00 | 41.54 | 42.00 | 43.72 | 38.93 |
| | $x=5$ | 55.50 | 65.33 | 48.73 | 47.47 | 56.78 | 67.86 | 74.00 | 51.76 | 61.09 |
| | $\Delta$ | 20.32 | 18.33 | 13.88 | 12.47 | 26.78 | 26.32 | 32.00 | 8.04 | 22.16 |
| **Vicuna-7b** | $x=0$ | 50.54 | 67.35 | 45.45 | 30.77 | 64.58 | 74.53 | 74.23 | 69.73 | 63.20 |
| | $x=5$ | 50.51 | 73.30 | 43.32 | 36.79 | 63.75 | 77.51 | 82.65 | 52.82 | 55.47 |
| | $\Delta$ | -0.03 | 5.95 | -2.13 | 6.02 | -0.83 | 2.98 | 8.42 | -16.91 | -7.73 |
| **Vicuna-13b** | $x=0$ | 69.19 | 83.92 | 49.24 | 57.07 | 79.50 | 81.03 | 87.76 | 68.50 | 65.77 |
| | $x=5$ | 72.45 | 83.82 | 59.78 | 59.69 | 80.25 | 79.59 | 88.66 | 72.86 | 65.31 |
| | $\Delta$ | 3.26 | -0.10 | 10.54 | 2.62 | 0.75 | -1.44 | 0.90 | 4.36 | -0.46 |
| **Vicuna-33b** | $x=0$ | 79.90 | 84.92 | 69.63 | 64.50 | 86.08 | 94.42 | 92.00 | 73.23 | 70.90 |
| | $x=5$ | 76.26 | 84.44 | 66.15 | 65.66 | 90.48 | 91.88 | 90.00 | 77.16 | 70.99 |
| | $\Delta$ | -3.64 | -0.48 | -3.48 | 1.16 | 4.40 | -2.54 | -2.00 | 3.93 | 0.09 |

## C.4 FEW-SHOT PROMPT

For Subtask1, we randomly sampled five different tools and, for each tool, randomly selected a corresponding query. The sampling method for Sub-task 2 was the same as Sub-task 1. For Subtask3, to balance different types of answers, we ensured that the ratio of answers being 'none' to answers being tool was either 2:3 or 3:2. Due to the task setup of Sub-task 4, we did not conduct few-shot experiments on Sub-task 4. This is because we have ensured that the 15 tools most similar to the ground-truth tool were not present in the tool list, and the number of multi-tool queries was limited, making it impossible to guarantee that each query could have five exemplars containing two-tool combinations.

Table 14: Comparison between human and LLMs. Model$_{\text{MAX}}$ is the best performance of LLMs and Model$_{\text{AVG}}$ is the average performance of eight LLMs.

| Sub-task | Similar | Scenario | Reliability | Multi-tool |
|---|---|---|---|---|
| **Model$_{\text{MAX}}$** | 73.46 | 79.51 | 50.35 | 88.28 |
| **Model$_{\text{AVG}}$** | 56.90 | 61.51 | 8.59 | 55.14 |
| **Human** | 86.00 | 91.00 | 96.00 | 66.00 |

## C.5 HUMAN EVALUATION

To investigate human performance in tool selection, we evaluated human abilities through question-naires.

Specifically, we mixed questions from four sub-tasks, asking participants to select 0 to 2 tools for each question. Each questionnaire comprised 10 or 15 questions, with participants making choices based on provided queries and candidate tools as options. We collected a total of 240 valid responses. The results of the human evaluation are presented in Table 14.

We observe a notable discrepancy between the CSR of humans and LLMs in sub-task 1, sub-task 2, and sub-task 3. Human CSR surpasses both the average and maximum CSR of LLMs. Notably, in sub-task 3, human performance reaches an impressive 96%, a stark contrast to the model's meager 9%. This discrepancy highlights the challenges LLMs face, particularly in addressing issues like hallucination, significantly impacting their reliability.

Moreover, in sub-task 4, human performance, while surpassing the average level of LLMs, falls short of reaching their maximum CSR. This implies that, when confronted with intricate language tasks, such as multiple-choice questions, LLMs still maintain a distinct advantage.

## D PROMPT TEMPLATE

### D.1 TOOLE DATASET GENERATION

We show the prompt templates of TOOLE dataset generation as follows:

---

**Direct diverse generation**

```
Here is a tool for ChatGPT, which can help it solve users'
requests better.  The description of this tool includes a
description of users and a description of ChatGPT.
The description of users:  {human description}
The description of ChatGPT: {model description}
Please give 10 examples where you would use this plugin to
answer a user's question and you should only tell me what
users will say.
Please ensure that the provided examples are distinct from
one another.  Feel free to employ various sentence styles,
such as instructions or requests, and vary the level of
detail as needed.
The format of your answer should be like:  1.  User:  [Your
Answer].  2.  User:  [Your Answer] ...
```

**Details diverse generation**

```
Here is a plugin designed to enhance ChatGPT's responsiveness
to users' needs.  ChatGPT only uses the plugin when it thinks
the tools will enhance its response.  Now, I would like you
to complete the following tasks:
I will provide you with a description of the plugin, and
based on that description, you need to provide five examples
of user inputs that would prompt ChatGPT to utilize the
plugin in order to enhance its responses for users.
Please ensure that your answers satisfy the following
conditions:
1.  Each example should be the first input in a new
conversation, without any prior context.
2.  The sentence should contain description information.
3.  Your answers should be as detailed as possible.
4.  Format your answers as follows:  1.  User:  [Answer], 2.
User:  [Answer], ...
5.  Utilizing this plugin has the potential to significantly
improve ChatGPT's ability to address users' requests.
The plugin includes descriptions for both users and ChatGPT.
The description for users is as follows:  {human
description}, and the description for ChatGPT is:  {model
description}.
```

**Keywords extraction for keywords generation**

```
Here is a tool for ChatGPT, which can help it solve users'
requests better.
The description of this tool includes a description of users
and a description of ChatGPT.
The description of users:  {human description}
The description of ChatGPT: {model description}.
Now your task is to give me five words or phrases to label
this tool.
These labels can be not mentioned in the description and
labels should be as short as possible.
The format of your answer should be 1.  [label 1], 2.  [label
2], ..., 5.  [label 5].
```

**Keywords generation**

```
Here is a tool for ChatGPT, which can help it solve users'
requests better.
The description of this tool includes a description of users
and a description of ChatGPT.
The description of users:  {human description}
The description of ChatGPT: {model description}
Now I will give you five labels of this tool and what you
need to do is give me five sentences the user will input to
ChatGPT when they may need the help of this tool.
Here are the labels:  {labels}
The format of your answer should be:  1.  [label 1]:
[sentence 1], 2.  [label 2]:  [sentence 2], ..., 5.  [label
5]:  [sentence 5].
```

---

**Emotional generation**

```
Here is a plugin designed to enhance ChatGPT's responsiveness
to users' needs.  ChatGPT only uses the plugin when it thinks
the tools will enhance its response.  Now, I would like you
to complete the following tasks:
I will provide you with a description of the plugin, and
based on that description, you need to provide five examples
of user inputs that would prompt ChatGPT to utilize the
plugin in order to enhance its responses for users.
Please ensure that your answers satisfy the following
conditions:
1.  Each example should be the first input in a new
conversation, without any prior context.
2.  The sentence should contain description information.
3.  The example should be in a {Emotion type} mood.
4.  Format your answers as follows:  1.  User:  [Answer], 2.
User:  [Answer], ...
5.  Utilizing this plugin has the potential to significantly
improve ChatGPT's ability to address users' requests.
The plugin includes descriptions for both users and ChatGPT.
The description for users is as follows:  {human
description}, and the description for ChatGPT is:  {model
description}.
```

---

**Multi-tool queries generation**

```
Now you are a query generation assistant.
Here are two tools for ChatGPT, which can help it solve
users' requests better.
The first tool is {first tool name}, its description is
"{first tool description}".
The second tool is {second tool name}, its description is
"{second tool description}".
Please give five examples where you would use these two tools
AT THE SAME TIME to answer a user's query and you should only
tell me what users will say.
Remember that the queries you give must be related to both
two tools!
Please ensure that the provided examples are distinct from
one another.  Feel free to employ various sentence styles,
such as instructions or requests, and vary the level of
detail as needed.
Remember that the 5 queries you generate should include both
parallel tool usage (two tools without any relationship)
and causal tool usage (one tool dependent on the result of
the other tool).  Also, the user's query cannot include the
specific name of the tool.
The format of your answer should be like:  1.  User:  [Your
Answer].  2.  User:  [Your Answer] ...
```

## D.2 PROMPT TEMPLATE OF EXPERIMENTS

We show the experimental prompt in this section including the Thought part (①) and the Action part (②).

**Thought (①) prompt**

```
You are an intelligent agent, and you need to constantly
be aware of your own limitations.  I will provide you with
a user's query, and you should assess, based on your own
capabilities, whether you need to use external tools to
better address the user's query.  Typically, there are four
reasons why you might need to use external tools:
[Reasons Begin]
{tool_reason}
[Reasons End]
Here is the user's query:
[User Query Begins]
{user_query}
[User Query Ends]
Based on the above query, if you think it's necessary to
use external tools, please respond with "yes"; otherwise,
respond with "no." Additionally, you should provide a brief
explanation for your answer.
```

**Action (②) prompt in the single-tool tasks**

```
You are a helpful AI assistant.  Your current task is to
choose the appropriate tool to solve the user's query based
on their question.  I will provide you with the user's
question and information about the tools.
If there is a tool in the list that is applicable to this
query, please return the name of the tool (you can only
choose one tool).  If there isn't, please return 'None.'
Additionally, you will need to support your answer with a
brief explanation.
User's Query:
[User's Query Start]
{user_query}
[User's Query End].
List of Tools with Names and Descriptions:
[List of Tools with Names and Descriptions Start]
{tool_list}
[List of Tools with Names and Descriptions End]
```

Table 15: Examples of tool usage awareness on ChatGPT.

| ID | Query | Answer |
|---|---|---|
| 1 | I need to remember to attend a meeting at 10 a.m. Can you set a reminder for me? | No, it is not necessary to use external tools for this query. As an intelligent agent, I have the capability to set reminders for users and can do so without the need for external tools. |
| 2 | Can you tell me more about the services offered by Magnetis? | No. As an AI language model, I have access to a vast amount of information on the internet and can gather information about services offered by Magnetis by searching relevant web pages and extracting information. Therefore, I do not need to use external tools to address this query. |
| 3 | My brother is a music lover. Can you find a gift related to music for him? | No. The query is straightforward and does not require external tools. The agent can search for music-related gift ideas using its own search capabilities. |
| 4 | What is the result of $3.0117^{3.82488}$? | No. As an AI language model, I can perform basic mathematical calculations, including exponentiation. I can directly calculate the result of 3.0117 raised to the power of 3.82488 without the need for external tools. |
| 5 | Find the four smallest perfect numbers. | Yes, it is necessary to use external tools to find the four smallest perfect numbers. |

```
Action (②) prompt in the multi-tool task

You are a helpful AI assistant.  Your current task is to
choose the appropriate tool to solve the user's query based
on their question.  I will provide you with the user's
question and information about the tools.
If there is a tool in the list that is applicable to this
query, please return the name of the tool (you can choose
two tools at most).  If there isn't, please return 'None.'
Additionally, you will need to support your answer with a
brief explanation.
User's Query:
[User's Query Start]
{user_query}
[User's Query End].
List of Tools with Names and Descriptions:
[List of Tools with Names and Descriptions Start]
{tool_list}
[List of Tools with Names and Descriptions End]
```

# E  FAILURE CASE STUDY

We show some failure cases of tool selection motivation in LLMs, as shown in Table 15 and Table 16. The following conclusions can be drawn from these cases: **(1) LLMs can be overconfident or lack confidence.** On one hand, LLMs demonstrate overconfidence in their capabilities, as exemplified by the first three instances in Table 15, where LLMs erroneously perceive themselves as having the capacity to offer reminders and conduct internet searches. On the other hand, LLMs display an unquestioning reliance on external tools, as evidenced by three instances in Table 16. Even for conventional language model tasks like sentiment classification and text generation, they prefer tool-based solutions. **(2) LLMs have conflicting perceptions of their own capabilities.** As demonstrated by the 4th, 5th, and 6th examples in Table 15, ChatGPT provides contradictory responses regarding the utilization of external tools when handling mathematical and logical queries. Based on the aforementioned findings, it is imperative to conduct additional research and training to enhance LLMs' proficiency in correctly utilizing tools.

Table 16: Examples of tool usage awareness on vicuna.

| ID | Query | Answer |
|---|---|---|
| 1 | Detect if the Reddit thread contains hate speech. r/worldnews
● Posted by u/madneon_ 3 hours ago
Russia 'borrows $13.6bn in largest ever debt issuance in a single day' to continue faltering war
● macross1984 · 3 hr. ago
Russia will become deadbeat. | Yes. To detect if the Reddit thread contains hate speech, external tools such as natural language processing (NLP) models or sentiment analysis algorithms may be required. These tools can analyze the text and identify any offensive or hateful language used in the comments. |
| 2 | Prove that every set is countable. | Yes. The query involves proving a mathematical statement, specifically about the countability of sets. As a language model, I can provide a general explanation or approach to solving this problem, but I cannot perform real-time calculations or generate formal proofs. To fully address the user's query, it would be beneficial to use external tools such as mathematical software (e.g., Mathematica, MATLAB) or consult with a human expert in the field of mathematics.. |
| 3 | Create a fun math question for children. | Yes, it is necessary to use external tools. The query requires the creation of a fun math question for children, which involves generating a specific type of content that may not fall within the capabilities of the language model. External tools or resources may be needed to generate appropriate and engaging math questions that are suitable for children. |

