# OpenReview forum: "MetaTool Benchmark for Large Language Models: Deciding Whether to Use Tools and Which to Use"
_ICLR.cc/2024/Conference — ICLR 2024 poster_

### Official Review · Reviewer_xJ5F · 2023-10-26

**Soundness:** 3 good
**Presentation:** 2 fair
**Contribution:** 3 good
**Rating:** 6
**Confidence:** 4

**Summary:**

The paper investigates whether models know when to use tools, and whether they know what tools to use.

The first task is “awareness of tool usage” and it’s a binary classification task. The authors construct the ToolE dataset for this task; the queries are generated using various prompting methods (“emotional generation,” “keyword generation,” “direct diverse generation,” “detailed generation).

The second task is “tool selection.” Four subtasks are proposed to evaluate tool selection: (1) Tool selection with similar choices: select the correct tool from a list containing similar tools. (2) Tool selection in specific scenarios (specialized scenarios). (3) Tool selection with possible reliability issues: given query and tool t, construct tool list such that the t is not in the tool list; assess if LLMs can avoid choosing tools that do not exist in the list. (4) Multi-tool selection.

The tasks seem to be done in a zero-shot manner based on the prompts in the appendix. For the first task, most models achieve ~random accuracy (except for ChatGPT and Vicuna which are better than the others – also potentially because these models are largest in the batch). Performance on the second task is unsatisfactory too.

**Strengths:**

Tool use is one of the most powerful ways to improve LLM performance, and there has been lots of recent interest (Toolformer, or even generating new tools).

It’s good that many of the examples shown in this paper are actually how humans may use an assistant in the future. So although the benchmark is artificial, the queries are often relevant to the real world.

**Weaknesses:**

The “awareness of tool usage” dataset contains positive and negative examples. Positive examples are the ones that require tools, and negative examples are the ones that can be directly solved by LLMs. For negative examples, the authors use three instruction tuning datasets like Commonsense QA and instruction datasets in LIMA. I’m a bit confused because a lot of those questions are quite difficult, and intuitively would definitely benefit from retrieval, for example. The reason LLMs can solve them may be because they appeared in the pretraining/fine-tuning datasets already, so LLMs remember the examples. So why are those instruction tuning datasets in the negative subset (if it wouldn’t harm if they search for info on the internet)?

Some answers (on whether LLMs need tools for a Q) in Table 12 and Table 13 are debatable in my opinion. I wonder how the authors define a reference answer (on whether LLMs need to use a tool) if with or without the tool, LLM can both solve the question. Similar issue for the second task (tool selection). What would be the human performance on these tasks?

Relevant to the above two: would it be more prudent to measure “whether LLMs know when to use tools / whether they can use tools” by actually performing tool use experiments – the metric would be the accuracy on the downstream task (e.g., whether the weather actually matches internet search, or whether some calculation actually equals the correct answer, or whether a 2-hour reminder is created in some simulator)?

The experiments are done zero-shot. I wonder if few-shot prompting (with chain of thought reasoning) can improve the performance (on both tasks) by a lot.

I wonder what the tool descriptions are. I don’t see detailed tool descriptions in the paper. I also don’t see any supplementary materials in the submission. In the tool description, is there info on *when* a *language model* should use the tool? If not, then I naturally don’t expect LLMs to do well on the tasks in this paper.

**Questions:**

Did you use the SFT-tuned & RLHF-tuned llama (which should be llama2-chat) or the plain pretrained llama (without SFT/RLHF finetuning)?

**Details Of Ethics Concerns:**

One of the specialized scenarios in the test set (the second task in Section 2.3.2) is called “housewife” covering topics like “discount, restaurant booking tool, product search, etc.” citing videos like “AI for women today, 10 genius ways housewives can use ChatGPT to save money.” I recognize there are many housewives but there seems to be a hint of prejudice and stereotyping there…? It would be nice to rephrase this category of examples. Perhaps I’m saying this because of my current physical location or social environment?

Another issue is the lack of descriptions of the tools. The results throughout the paper rely on them, but they are not available.

UPDATE: These two concerns are addressed by the authors as of 11/23.

---

> ### Author Response · Authors · 2023-11-18
> **Thank you for your valuable feedback - Response (1)**
>
> Thank you very much for your feedback. We will explain step by step to clarify your doubts and thus, as much as possible, eliminate any misunderstanding about your contribution to the article.
>
> Q: I’m a bit confused because many of those questions are quite difficult, and intuitively they would definitely benefit from retrieval. The reason LLMs can solve them might be because they appeared in the pretraining/fine-tuning datasets already, so LLMs remember the examples. So why are those instruction-tuning datasets in the negative subset (if it wouldn’t harm if they search for info on the internet)?
>
> A: Firstly, it is crucial to understand the significance of incorporating negative samples.
>
> The purpose of introducing negative samples is to ensure the fairness of the tool usage awareness evaluation. Without them, models might excessively rely on tools for all types of queries, which is not the ideal performance we expect from LLMs. For example, if LLMs use tools in all situations, including answering simple mathematical questions (like "5 * 8 = ?"), they would be no different from standard search engines, losing their characteristics as advanced language processing systems. Therefore, introducing negative samples helps make task assessments more comprehensive and balanced.
>
> Secondly, we need to redefine the fundamental principles of LLMs using tools. It is reasonable to use tools when queries (instructions) exceed the processing capabilities of LLMs. For instance, even though basic operations (such as addition, subtraction, multiplication, and division) can be completed by tools with possibly very accurate results, if LLMs can handle these problems independently, then using tools becomes redundant. Otherwise, LLMs' advanced natural language processing and generative capabilities would not be fully demonstrated, and further research and enhancement of their computational and reasoning abilities would be meaningless. Therefore, as we proposed in Appendix A.4, LLMs should consider using tools only in the following four situations: when real-time data or external databases are needed (A), for processing special inputs/outputs, such as multimodal inputs (B), for handling domain-specific tasks, such as complex computational tasks (C), and in scenarios involving user interaction, like playing games (D).
>
> We will introduce how we select positive and negative samples in the next question.

---

> ### Author Response · Authors · 2023-11-18
> **Response (2)**
>
> Q: I wonder how the authors define a reference answer (on whether LLMs need to use a tool) if with or without the tool, LLM can both solve the question.
>
> A: We are sorry for your confusion. To better understand the Tool Usage Awareness dataset, we will describe how we selected positive and negative samples for the dataset. Firstly, we categorized user queries into three types:
> The first type is "Queries must be solved by tools" (positive), such as multimodal input and real-time information retrieval.
> The second type is "Queries can be solved well by all LLMs" (negative), such as telling jokes, basic conversational functions, sentiment classification, and other basic NLP tasks.
> The third type represents the middle ground between the first and second types of user queries, i.e.,  queries we hope LLMs can solve but currently cannot, such as complex calculations, long text summarization, and information extraction.
>
> With the aforementioned types of user queries, we classified user queries into positive or negative samples using human evaluation and model checking.
>
> (1) Human evaluation: For the first and second types, we determined the samples through unanimous agreement from two human experts and referenced the four reasons in Appendix A.4 for selection.
>
> (2) Model Checking: Regarding the third type of user query, our objective is to find those within this category that can be solved well by all LLMs (classified as negative) and those that none of the LLMs can solve (classified as positive). We discarded user queries that only a portion of the LLMs can solve them. We conduct validation in the following two steps:
>
> We initially input the queries into GPT-4. Since GPT-4 currently has the best performance in terms of utility, if GPT-4 declines to answer (i.e., unable to solve the problem or refuse to answer), we classify it as a positive query.
>
> Following the above operation, if the query is not classified as positive, we conducted inference on eight LLMs and evaluated the answers through two human experts. If all output from LLMs solves the problem well, we classify them as negative queries.
>
> **We have added detailed information in Appendix B (blue text).**

---

> ### Author Response · Authors · 2023-11-18
> **Response (3)**
>
> Q: Relevant to the above two: would it be more prudent to measure “whether LLMs know when to use tools / whether they can use tools” by actually performing tool use experiments – the metric would be the accuracy on the downstream task (e.g., whether the weather actually matches internet search, or whether some calculation actually equals the correct answer, or whether a 2-hour reminder is created in some simulator)?
>
> A: Thank you very much for your suggestion. However, we would like to emphasize that the primary focus of this paper is to assess whether LLMs exhibit tool usage awareness (knowing when to use tools) and whether they can choose the correct tools. Our emphasis is on the inherent capabilities of LLMs. In contrast, the metric you mentioned, "the accuracy on the downstream task," largely depends on the settings and capabilities of the tools themselves, which is not closely related to the central theme of this paper. Figure 1 in the paper illustrates our key points, highlighting 1 and 2 (Thought and Action), rather than 3 and 4. Furthermore, we need to reiterate the contributions of this paper:
>
> 1. We introduce ToolE, a comprehensive dataset that encompasses a wide range of 21,127 user queries, with both single-tool and multi-tool queries. Different from the previous single-method generation, these queries are generated using various prompting methods, including emotional generation, keyword generation, direct diverse generation, and detailed generation. Moreover, to address the challenge of overlapping tool functionality, we undertake tool merging and decomposition.
> 2. Evaluation on awareness of tool usage and tool selection. We construct a test set to evaluate the awareness of tool usage based on ToolE and existing instruction datasets. Moreover, we formulate four distinct tasks to evaluate the tool selection ability of LLMs. These tasks are thoughtfully designed to assess semantic comprehension, adaptability, reliability, and inferential capability, namely tool selection with similar choices, tool selection in specific scenario, tool selection with possible reliability issues, and multi-tool selection.
> 3. We rigorously evaluate the performance of eight well-known LLMs. We have observed that most LLMs struggle to recognize their capability boundaries and lack a good awareness of tool usage. Regarding tool selection, we find that while LLMs possess basic tool selection capabilities, the tool selection of most LLMs remains unreliable, with noticeable variations in performance across different daily scenarios. Moreover, the error analysis indicates there is still room for improvement in tool selection. Finally, by analysis the tool description, we gained two insights for tool developer.

---

> ### Author Response · Authors · 2023-11-18
> **Response (4)**
>
> Q: The experiments are done zero-shot. I wonder if few-shot prompting (with chain of thought reasoning) can improve the performance (on both tasks) by a lot.
>
> A: We supplemented the few-shot prompts experiments about tool usage awareness and the first three subtasks of tool selection, as outlined in Appendix C.4 of our experimental setup. Due to the lack of annotated CoT data, we had the LLMs generate a segment of reasoning to explain their answers, achieving a similar purpose of CoT.
>
> **Tool Usage Awareness:**
> | Metrics   | ChatGPT | ChatGLM2 | Llama2-7b | Llama2-13b | Vicuna-7b | Vicuna-13b | Vicuna-33b | Koala-13b |
> | --------- | ------- | -------- | --------- | ---------- | --------- | ---------- | ---------- | --------- |
> | **$x=0$** |         |          |           |            |           |            |            |           |
> | Acc.      | 74.85   | 54.56    | 53.88     | 51.94      | 52.43     | 61.17      | 65.05      | 54.95     |
> | Pre.      | 69.63   | 61.90    | 52.71     | 80.00      | 74.51     | 82.49      | 79.92      | 55.53     |
> | Rec.      | 88.16   | 40.39    | 84.85     | 6.21       | 7.38      | 28.35      | 40.19      | 52.62     |
> | F1        | 77.81   | 48.88    | 65.03     | 11.53      | 13.43     | 42.2       | 53.49      | 54.04     |
> | **$x=5$** |         |          |           |            |           |            |            |           |
> | Acc.      | 79.71   | 56.02    | 50.39     | 57.86      | 56.12     | 55.15      | 61.55      | 54.66     |
> | Pre.      | 73.98   | 64.32    | 50.55     | 63.31      | 56.49     | 52.92      | 57.85      | 52.61     |
> | Rec.      | 91.65   | 51.46    | 99.03     | 47.57      | 54.95     | 96.89      | 85.83      | 97.86     |
> | F1        | 81.87   | 57.17    | 66.93     | 54.32      | 55.71     | 68.45      | 69.12      | 68.43     |
> | F1 Δ      | 4.06 ↑  | 8.92 ↑   | 1.90 ↑    | 42.79 ↑    | 42.28 ↑   | 26.25 ↑    | 15.63 ↑    | 14.39 ↑   |
>
> **Tool selection: Sub-task 1 & Sub-task 2:**
>
> | Metric          | ChatGLM2 | ChatGPT | Llama2-7b | Llama2-13b | Vicuna-7b | Vicuna-13b | Vicuna-33b | Koala-13b |
> | --------------- | -------- | ------- | --------- | ---------- | --------- | ---------- | ---------- | --------- |
> | **Similar**     |          |         |           |            |           |            |            |           |
> | $x=0$           | 54.17    | 69.05   | 45.95     | 44.06      | 73.46     | 58.23      | 53.96      | 56.34     |
> | $x=5$           | 57.44    | 72.94   | 51.12     | 49.85      | 63.67     | 63.15      | 60.54      | 60.85     |
> | $\Delta$        | 3.27 ↑   | 3.89 ↑  | 5.17 ↑    | 5.79 ↑     | -9.79 ↓   | 4.92 ↑     | 6.58 ↑     | 4.51 ↑    |
> | **Reliability** |          |         |           |            |           |            |            |           |
> | $x=0$           | 6.63     | 50.35   | 0.90      | 2.31       | 1.50      | 2.51       | 2.81       | 1.70      |
> | $x=5$           | 15.68    | 78.49   | 2.51      | 5.93       | 1.81      | 3.42       | 3.11       | 5.83      |
> | $\Delta$        | 9.05 ↑   | 28.14 ↑ | 1.61 ↑    | 3.62 ↑     | 0.31 ↑    | 0.91 ↑     | 0.30 ↑     | 4.13 ↑    |

---

> ### Author Response · Authors · 2023-11-18
> **Response (5)**
>
> (In continuation of the previous answer)
>
> **Sub-task 3:**
> | Scenario       | Finan.  | Home.   | Soft.   | Stud.   | Artis.  | Elders  | Top 5   | Top 10   | Top 15  |
> | -------------- | ------- | ------- | ------- | ------- | ------- | ------- | ------- | -------- | ------- |
> | **ChatGLM2**   |         |         |         |         |         |         |         |          |         |
> | $x=0$          | 46.70   | 57.59   | 28.50   | 40.31   | 42.78   | 68.42   | 82.29   | 56.19    | 48.07   |
> | $x=5$          | 62.57   | 62.94   | 41.53   | 34.97   | 45.64   | 69.66   | 86.17   | 57.22    | 50.54   |
> | $\Delta$       | 15.87 ↑ | 5.35 ↑  | 13.03 ↑ | -5.34 ↓ | 2.86 ↑  | 1.24 ↑  | 3.88 ↑  | 1.03 ↑   | 2.47 ↑  |
> | **ChatGPT**    |         |         |         |         |         |         |         |          |         |
> | $x=0$          | 69.70   | 82.23   | 76.70   | 66.67   | 80.00   | 83.76   | 88.89   | 77.39    | 75.08   |
> | $x=5$          | 74.24   | 86.73   | 69.11   | 72.45   | 87.00   | 89.45   | 91.00   | 80.50    | 77.33   |
> | $\Delta$       | 4.54 ↑  | 4.50 ↑  | -7.59 ↓ | 5.78 ↑  | 7.00 ↑  | 5.69 ↑  | 2.11 ↑  | 3.11 ↑   | 2.25 ↑  |
> | **Koala-13b**  |         |         |         |         |         |         |         |          |         |
> | $x=0$          | 59.32   | 69.47   | 47.69   | 43.75   | 73.58   | 77.78   | 86.42   | 55.56    | 52.98   |
> | $x=5$          | 51.83   | 61.85   | 46.99   | 53.44   | 64.38   | 82.35   | 81.71   | 69.82    | 52.59   |
> | $\Delta$       | -7.49 ↓ | -7.62 ↓ | -0.70 ↓ | 9.69 ↑  | -9.20 ↓ | 4.57 ↑  | -4.71 ↓ | 14.26 ↑  | -0.39 ↓ |
> | **Llama2-7b**  |         |         |         |         |         |         |         |          |         |
> | $x=0$          | 52.04   | 57.07   | 30.00   | 32.99   | 59.09   | 57.14   | 65.66   | 45.73    | 44.86   |
> | $x=5$          | 64.97   | 62.12   | 42.35   | 40.00   | 69.00   | 76.17   | 67.68   | 47.24    | 54.08   |
> | $\Delta$       | 12.93 ↑ | 5.05 ↑  | 12.35 ↑ | 7.01 ↑  | 9.91 ↑  | 19.03 ↑ | 2.02 ↑  | 1.51 ↑   | 9.22 ↑  |
> | **Llama2-13b** |         |         |         |         |         |         |         |          |         |
> | $x=0$          | 35.18   | 47.00   | 34.85   | 35.00   | 30.00   | 41.54   | 42.00   | 43.72    | 38.93   |
> | $x=5$          | 55.50   | 65.33   | 48.73   | 47.47   | 56.78   | 67.86   | 74.00   | 51.76    | 61.09   |
> | $\Delta$       | 20.32 ↑ | 18.33 ↑ | 13.88 ↑ | 12.47 ↑ | 26.78 ↑ | 26.32 ↑ | 32.00 ↑ | 8.04 ↑   | 22.16 ↑ |
> | **Vicuna-7b**  |         |         |         |         |         |         |         |          |         |
> | $x=0$          | 50.54   | 67.35   | 45.45   | 30.77   | 64.58   | 74.53   | 74.23   | 69.73    | 63.20   |
> | $x=5$          | 50.51   | 73.30   | 43.32   | 36.79   | 63.75   | 77.51   | 82.65   | 52.82    | 55.47   |
> | $\Delta$       | -0.03 ↓ | 5.95 ↑  | -2.13 ↓ | 6.02 ↑  | -0.83 ↓ | 2.98 ↑  | 8.42 ↑  | -16.91 ↓ | -7.73 ↓ |
> | **Vicuna-13b** |         |         |         |         |         |         |         |          |         |
> | $x=0$          | 69.19   | 83.92   | 49.24   | 57.07   | 79.50   | 81.03   | 87.76   | 68.50    | 65.77   |
> | $x=5$          | 72.45   | 83.82   | 59.78   | 59.69   | 80.25   | 79.59   | 88.66   | 72.86    | 65.31   |
> | $\Delta$       | 3.26 ↑  | -0.10 ↓ | 10.54 ↑ | 2.62 ↑  | 0.75 ↑  | -1.44 ↓ | 0.90 ↑  | 4.36 ↑   | -0.46 ↓ |
> | **Vicuna-33b** |         |         |         |         |         |         |         |          |         |
> | $x=0$          | 79.90   | 84.92   | 69.63   | 64.50   | 86.08   | 94.42   | 92.00   | 73.23    | 70.90   |
> | $x=5$          | 76.26   | 84.44   | 66.15   | 65.66   | 90.48   | 91.88   | 90.00   | 77.16    | 70.99   |
> | $\Delta$       | -3.64 ↓ | -0.48 ↓ | -3.48 ↓ | 1.16 ↑  | 4.40 ↑  | -2.54 ↓ | -2.00 ↓ | 3.93 ↑   | 0.09 ↑  |
>
> We observed that, in the tool usage awareness task, the accuracy of LLMs showed both improvements and declines, but the F1 Score increased. This indicates a significant enhancement of few-shot prompts on the tool usage awareness task.
>
> Additionally, in the tool selection task, for subtask 1, we found that, apart from Vicuna-7b (possibly due to the model's poor robustness and high sensitivity), few-shot prompts improved the performance of other LLMs. However, in subtask 2, this improvement was limited for most models. ChatGPT achieved a remarkable 28.14% improvement, and ChatGLM2 also saw a 9.05% improvement. The Vicuna series and LLama2 series models showed less than 5% improvements. In subtask 3, few-shot prompts enhanced the performance of most LLMs, especially for the LLama2 series models, which demonstrated improvements in CSR across various scenario scenarios.

---

> ### Author Response · Authors · 2023-11-18
> **Response (6)**
>
> Q: What would be the human performance on these tasks?
>
> A: We evaluated human abilities through questionnaires to investigate human performance in tool selection. Specifically, we mixed questions from four sub-tasks, asking participants to select 0 to 2 tools for each question. Each questionnaire comprised 10 or 15 questions, with participants making choices based on provided queries and candidate tools as options. We collected a total of 340 valid responses due to the time limitation.
>
> Model$_{Max}$ is the best performance of LLMs.
>
> Model$_{Avg}$ is the average performance of eight LLMs.
>
> | **Sub-task** | **Similar** | **Scenario** | **Reliability** | **Multi-tool** |
> | ------------- | ----------- | ----------- |--------------- | -------------- |
> | Model$_\text{Max}$ | 73.46  |  79.51   | 50.35           | 88.28          |
> | Model$_\text{Avg}$ | 56.90  |   61.51  | 8.59            | 55.14          |
> | Human              | 86.00  |  91.00     | 96.00           | 66.00          |
>
> We observe a notable discrepancy between the CSR of humans and LLMs in sub-task 1, sub-task 2, and sub-task 3. Human CSR surpasses both the average and maximum CSR of LLMs. In sub-task 3, human performance reaches an impressive 96%, starkly contrasting to the model's meager 9%. This discrepancy highlights LLMs' challenges, particularly in addressing issues like hallucination, significantly impacting their reliability.
>
> Moreover, in sub-task 4, while surpassing the average level of LLMs, human performance falls short of reaching their maximum CSR. This implies that LLMs still maintain a distinct advantage when confronted with intricate language tasks, such as multiple-choice questions.
>
> **We have added detailed information in Appendix C.5 (blue text).**

---

> ### Author Response · Authors · 2023-11-18
> **Response (7)**
>
> Q: Did you use the SFT-tuned & RLHF-tuned llama (which should be llama2-chat) or the plain pretrained llama (without SFT/RLHF fine-tuning)?
>
> A: Thanks a lot for your careful consideration. We used the chat version and have added details in the experiment setting.
>
> ---
>
> Q: I wonder what the tool descriptions are. I don’t see detailed tool descriptions in the paper. I also don’t see any supplementary materials in the submission.
>
> A: According to your suggestion, we have uploaded the description information of the tool to the supplementary materials. The developers themselves write the descriptions of these tools and you can also find them in the OpenAI plugin store.
>
> ---
>
> Q: Ethics Concerns. One of the specialized scenarios in the test set (the second task in Section 2.3.2) is called “housewife” covering topics like “discount, restaurant booking tool, product search, etc.” citing videos like “AI for women today, 10 genius ways housewives can use ChatGPT to save money.” I recognize many housewives but there seems to be a hint of prejudice and stereotyping there…? It would be nice to rephrase this category of examples. Perhaps I’m saying this because of my current physical location or social environment?
>
> A: We sincerely apologize for using the term "housewife" in our paper, which may carry a biased connotation, and we deeply appreciate your feedback. Based on your suggestion, we realize that biased terminology might inadvertently reinforce inappropriate stereotypes. Therefore, in the relevant sections of our paper, we plan to replace "housewife" with "home manager" to more fairly and broadly represent those who manage daily household affairs and budgets. This change will be reflected in our revised manuscript. We hope this modification will more accurately convey the intent of our research while avoiding any potential bias.
>
> Finally, heartfelt thanks for the comments you provided on this paper. We hope that our responses can help resolve any confusion you may have.

---

> ### Author Response · Authors · 2023-11-22
>
> Dear Reviewer,
>
> Thank you for your invaluable assistance and support. Given the constraints of time, we wish to ensure that our responses have effectively addressed any concerns you may have had. If there are still lingering issues, please feel free to inform us. We eagerly anticipate your additional feedback and hope that, if all your primary concerns have been resolved, you may reconsider raising your score.
>
> Once again, we appreciate your time and effort in reviewing our paper.

---

> ### Author Response · Authors · 2023-11-22
> **Thanks for your comment.**
>
> Dear reviewer xJ5F,
>
> We have addressed all concerns raised in the initial review comprehensively, including the incorporation of additional experiments and detailed explanations. Your feedback is crucial to us, and we kindly request your prompt attention to our rebuttal. **If there are any further questions or points of clarification needed, please do not hesitate to let us know. Your timely response would be greatly appreciated.**

---

> ### Comment · Reviewer_xJ5F · 2023-11-23
>
> I really appreciate the detailed response. I'm still going over the results and other reviewers' comments, and I do plan on increasing the score in the next day or two.

---

> > ### Author Response · Authors · 2023-11-23
> > **Thanks for your feedback.**
> >
> > Dear Reviewer,
> >
> > We sincerely appreciate the time and effort you have invested in evaluating our work and are pleased to hear that you appreciated the detailed response in our rebuttal.
> >
> > We remain open to further feedback and are willing to make necessary improvements or clarifications to our research. We look forward to your final score and respect any decision you make. Thank you again for your attention and efforts towards our work.

---

### Official Review · Reviewer_hidW · 2023-11-01

**Soundness:** 4 excellent
**Presentation:** 3 good
**Contribution:** 3 good
**Rating:** 6
**Confidence:** 4

**Summary:**

The METATOOL Benchmark paper introduces a new benchmark designed to evaluate the tool usage awareness and selection abilities of large language models (LLMs). The TOOLE dataset contains various user queries that prompt LLMs to use tools, and the benchmark includes tasks for both tool usage awareness and tool selection. The paper presents the results of experiments with several LLMs on the benchmark, and analyzes the performance of the models on different subtasks. The authors also provide insights into the factors that influence tool selection, and discuss potential real-world applications for LLMs with strong tool usage awareness and selection abilities. Overall, the paper's contributions include the development of a new benchmark for evaluating LLMs, insights into the factors that influence tool selection, and a framework for analyzing the performance of LLMs on tool usage tasks.

**Strengths:**

- The paper is well-written and easy-to-follow. The visualizations are clear and can help readers easily understand the task definitions, model performances, and framework.
- The paper is highly original in its approach to evaluating the tool usage awareness and selection abilities of large language models. The authors introduce a new benchmark, the TOOLE dataset, which includes a wide range of user queries generated using various prompting methods.
- The authors provide detailed descriptions of the TOOLE dataset and the four subtasks in tool selection, and they analyze the performance of several LLMs on the benchmark. The paper's results are presented clearly and are supported by statistical analysis.
- The conclusions are very insightful: the more detailed the description, the more efficient tool selection.

**Weaknesses:**

- I think the comparison with existing benchmarks still needs polishing. To the best of my knowledge, I have found these places to be potentially incorrect: (1) API-Bank (https://arxiv.org/pdf/2304.08244v1.pdf) has included the task ① of determining whether LLMs need to leverage external tools or not (level-1 description in Section 1 Page 2. Thus, it might be incorrect to describe API-Bank as only focusing on tasks ③④; (2) ToolBench (https://arxiv.org/abs/2305.16504) also involves the task ②, requiring the retriever to retrieve the most relevant tools. Thus, it is also incorrect to describe the ToolBench as a benchmark that only focuses on tasks ③④; (3) There are also some benchmarks that are missing in the table, like ToolQA (https://arxiv.org/pdf/2306.13304.pdf). In summary, the authors better clarify the differences compared with these existing benchmarks more clearly.
- Most statements in the experiments are not well explained. I think the authors should focus more on the potential reasons and analysis behind these observations.
- I am a little bit confused about the second conclusion you have obtained in Section 3.3. Can you provide more details about the comparison between generated tool descriptions and those provided by tool developers. Like, if we rewrite all the tool descriptions by tool developers with ChatGPT, will we observe a performance gain?

**Questions:**

See Weaknesses

---

> ### Author Response · Authors · 2023-11-16
> **Thank you for your valuable feedbacks.**
>
> Thank you very much for your suggestion! We apologize for the confusion caused, and next, we will gradually explain your concerns, to give you a clearer understanding of your contributions to the work:
>
> Q: I think the comparison with existing benchmarks still needs polishing.
>
> A: First, we acknowledge that there are some deficiencies in the comparison with benchmarks, which may be due to differences in understanding from various perspectives of different persons. Based on the issues identified after your careful review, we have revisited the related literature and engaged in in-depth discussions. We have made modifications to the comparison as per your request.

---

> ### Author Response · Authors · 2023-11-16
> **Response (2)**
>
> Q: Most statements in the experiments are not well explained.
>
> A: To address your concern, we have carefully reviewed the analysis of our experimental results and found that we indeed did not effectively analyze the experimental outcomes. We have implemented the following improvements:
>
> 1. Additional Experiments: We added **few-shot prompt experiments** for three subtasks of the tool usage awareness task and the tool selection task; we introduced **a new prompt template in multi-tool selection** to specify the number of tools returned by LLMs.
> 2. More In-Depth Analysis of Experimental Results: For both the original and the additional experiments mentioned above, we conducted a comprehensive and profound analysis of our experimental results. We have reached the following conclusions:
>
> (1) **Even under few-shot prompts, the majority of LLMs still perform poorly in tool usage awareness.** In Table 2, we observe that under the zero-shot prompt, only ChatGPT has both accuracy and F1 score exceeding 70\%, while the performance of other models is relatively poor, with the F1 score of llama2-13b being only 11.53%. Under the five-shot prompt, some models show significant improvement in F1 scores, for example, llama2-13b increased by 42.79%, and vicuna-7b by 42.28\%. This indicates that though few-shot learning generally improves the performance of LLMs in tool usage awareness, they still lack sufficient tool usage awareness.
>
>
> (2) **When selecting similar tools, there is a significant performance disparity among existing LLMs, and the improvement brought by few-shot prompts is limited.** Table 3 shows that under the zero-shot prompts, the best-performing LLM is Vicuna-7b, with nearly a 30\% difference compared to the worst-performing Llama2-13b. The gap between the best-performing ChatGPT and the worst-performing Llama2-13b still exceeds 20% under 5-shot prompts. Additionally, the maximum improvement brought by 5-shot prompts does not exceed 7%. Moreover, the performance of Vicuna-7b even declined by 10\% under the five-shot condition, suggesting a potential bias in its 0-shot performance, which reflects either a lack of robustness or over-sensitivity of the model.
>
> (3) **LLMs still face serious challenges in dealing with reliability issues, such as reducing hallucinations.** As seen from Table 3, although few-shot prompts improve the performance of all LLMs, the CSR of most LLMs remains below 20\%. We find that LLMs sometimes fabricate non-existent tools, a severe hallucination issue that has a significant impact on LLM-based agents. Additionally, the potential sycophancy of LLMs may lead them to avoid returning a "none" answer, instead choosing irrelevant tools to respond to users.
>
> (4) **LLMs perform poorly in processing long texts.** From Figure 4, we can see that the CSR of almost all LLMs decreases as the length of the tool list increases, especially in the range from the top 5 to the top 10. This indicates that LLMs still need improvement in understanding long texts. **LLMs exhibit imbalances and biases in tool selection across different scenarios.** For example, in Figure 5, LLMs generally have a higher CSR in tool selections related to the elderly and artists & designers, while their CSR is lowest for tools related to students. This means that developers still need to enhance the generalization capabilities of LLMs. At the same time, for downstream applications, it is best to choose suitable LLMs based on different applied fields.
>
> (5) **There are significant performance differences among LLMs in multi-tool selection.** As shown in Table 4, ChatGPT, the top-performing model, outperforms ChatGLM2, the worst-performing model, by nearly 70%, highlighting the variability in the capabilities of different language models for this task. Furthermore, the most common error made by the models is omitting tool selection, such as in the case of Vicuna-33b, which only selected one tool in 48.49% of cases. Moreover, **several LLMs overly rely on the explicitly specified number of tools they should select in the prompts.** When explicitly instructed to return two tools, Vicuna-33b's correct selection rate increased to over 90%, and Vicuna-7b also improved by over 20%. This indicates that these LLMs still possess good multi-tool selection capabilities but require prior knowledge, which makes it challenging to apply in LLM-based agents.
>
> **The detailed analysis for the above conclusions is updated in the PDF document, marked in blue font.**

---

> ### Author Response · Authors · 2023-11-16
> **Response 3**
>
> Q: I am a little bit confused about the second conclusion you have obtained in Section 3.3.
>
> A: Thank you very much for your suggestion and for expressing such detailed concerns! To address your concerns, we have rewritten all tool descriptions using GPT-4 and LLama2-70b and examined the performance of LLMs on subtask 1 of tool selection. The performance is as follows (we have also displayed this result in Figure 6 in the PDF):
>
> | Description | ChatGLM2 | ChatGPT | Llama2-7b | Llama2-13b | Vicuna-7b | Vicuna-13b | Vicuna-33b | Koala-13b | Average |
> | --- | --- | --- | --- | --- | --- | --- | --- | --- | --- |
> | Llama2-70b | 1.44 | 5.42 | -6.12 | 7.83 | 3.05 | 2.58 | -4.37 | -5.02 | 0.6 |
> | GPT-4 | -10.85 | -0.44 | -5.77 | -1.23 | 2.38 | 7.24 | 14.49 | -3.31 | 0.32 |
>
> We observed that different rewritten LLMs yielded varying benefits for different groups. For instance, descriptions rewritten by Llama2-70b resulted in a 7.83% improvement for llama2-13b but did not significantly enhance the performance of the Vicuna series models. In contrast, descriptions rewritten by GPT-4 caused a sharp decline in the performance of ChatGLM and Llama2 series, while significantly boosting the Vicuna series, possibly due to the Vicuna series' training corpus being largely sourced from ShareGPT.
>
> For a more accurate expression, we have modified the original statement "descriptions generated by chatgpt are better than those written by the tool developers themselves" to "we strongly recommend that tool developers choose an appropriate rewrite model for generating new descriptions based on the downstream LLM the tool will apply to.“

---

> ### Comment · Reviewer_hidW · 2023-11-20
> **Official Comment from Reviewer hidW**
>
> Thank you for your efforts during the rebuttal stage. My concerns have been resolved. I will maintain my positive score and vote for acceptance. Also, I noticed that all the rebuttals are not available to the public. Please double-check.

---

> > ### Author Response · Authors · 2023-11-20
> > **Thanks a lot for your comment.**
> >
> > Thanks a lot for your comment. We have made our rebuttals available to the public.

---

### Official Review · Reviewer_sQgt · 2023-11-01

**Soundness:** 3 good
**Presentation:** 3 good
**Contribution:** 3 good
**Rating:** 8
**Confidence:** 4

**Summary:**

This works studies the ability of LLMs to (1) decide whether to use tools and (2) which tools to use. They introduce a benchmark with over 20,000 prompts that trigger single- or multi-tool usage. Unlike recent related benchmarks, they study the reliability of nine LLMs at tool selection between similar choices, across challenging and diverse scenarios, and when multiple tools are needed. To generate diverse scenario prompts, the authors use algorithms that apply prompting and tool merging & decomposition. Through multiple studies and error analyses, the authors characterize ways in which most LLMs are inconsistent and poorly calibrated at tool usage.

The authors emphasize tool usage as a concrete way to test hallucination (does the LM know when it needs help from a tool? does the LM know which tool to use?) and sycophancy (does the LM know when to defer to tools despite context in the prompt?). The work also makes interesting connections between recommendation (or retrieval) and tool usage from an evaluation standpoint.

**Strengths:**

1. This benchmark tackles an area with much hype, and offers a much-needed challenging "agent" benchmark.

2. The authors thoughtfully construct MetaTool, following an extensive consideration of the desiderata (Sec 2). This work could be part of making "LLM general-purpose agents" a more systematic and empirical area.

3. The construction of the dataset is relatively novel. The authors collect 390 tools from OpenAI plugins. They merge (or decompose) tools to reduce benchmark ambiguity.  To construct the queries for evaluation, they prompt OpenAI models in a set of four pipelines. They also generate multi-tool queries from pairs of tools. All queries in the dataset were checked by a human.

4. The definition of the four tasks and the evaluation analyses conducted (3.2, 3.3) are quite rich.

**Weaknesses:**

1. Section 2 is rich in substance but the presentation is disorderly. There's probably at least 2 different sections that should be there, like "Task Formulation & Desiderata" first then "Dataset Generation" second, but these two considerations are mixed up repeatedly in the current presentation. I'm happy to change my mind on this if the authors would like to defend the current organization.

2. It's unclear (and seemingly undiscussed) how fundamentally the generation of the test queries with LLMs biases the evaluation for/against particular models, or reduce the true diversity of these queries.

**Questions:**

Questions and comments:

1. What kind of dev vs. test guidance do you recommend for this benchmark? Is it meant to be a truly "blind zero-shot, no tuning" dataset? What did the authors do for tuning? Did they use the prompt unchanged for each LLM? (Having no dev set is completely reasonable in principle, but practical considerations tell us that much of the community will, in effect, reuse the test set so often it becomes a dev set. I recommend having at least a few examples explicitly for tuning.)

2. Which Llama2 models were used, vanilla or chat? I presume it's chat. It may be good to include more explicit information in the main text. You find that "the worst-performing model, Llama2-13b, has an F1 Score of only 11.53%". This is surprisingly low. Is this some kind of unintended interaction between the chat format or some other minor detail and the task?

3. Is the word "task" is overloaded in Sec 2.3, where it refers both to the two major tasks "2.3.1" and "2.3.2", and to the four sub-tasks of "2"? Maybe rename the four tool selection tests to "sub-tasks".

---

> ### Author Response · Authors · 2023-11-12
> **Thank you for your valuable feedbacks.**
>
> We greatly appreciate your feedback! We will address your concerns one by one:
>
> Q: Section 2 is rich in substance but the presentation is disorderly.
>
> A: Thanks for your feedback. We acknowledge that we did not organize the structure of this section well. However, swapping the positions of dataset generation and task formulation might not be suitable. This is mainly because many definitions in task formulation rely on the premise that the reader is already clear about the structure of the dataset. Therefore, placing dataset generation after task formulation might cause more confusion for the reader.
>
> However, we appreciate your expression of confusion, so we have added a brief introduction at the beginning of Section 2 (already updated in the PDF in blue font) to help readers better understand the structure of the section.
>
> ---
>
> Q: It's unclear (and seemingly undiscussed) how fundamentally the generation of the test queries with LLMs biases the evaluation for/against particular models or reduces the true diversity of these queries.
>
> A: Thank you for pointing out this issue! We acknowledge this is a concern and still exists in the current LLMs-generated dataset.
>
> Specifically, we attempted to use various models to generate a mixed dataset, such as Llama2-70b. However, in the actual generation process, we found that the diversity of samples generated by Llama2-70b was lacking, and the quality of generated samples was low (e.g., not relevant to the tool's description information). However, MT-Bench [1] indicated that ChatGPT has a low bias on self-enhancement. Therefore, we believe ChatGPT does not strongly favor its own generated content, making it the least biased choice for generating data while ensuring data quality.
>
> ---
>
> Q: What kind of dev vs. test guidance do you recommend for this benchmark?
>
> A: Your suggestion is forward-thinking. Inspired by your advice, we have started constructing dev. Specifically, we are going to create a fine-tune dataset for tool usage awareness. We plan to use all positive samples (all LLMs must use the tool to answer questions) as input to GPT-4 and obtain answers (output should be a refusal to answer, e.g., “Sorry, I cannot answer this…”). We intend to use positive samples and their corresponding answers as the fine-tune dataset to improve LLMs' tool usage awareness. This ensures LLMs can accurately identify inputs beyond their ability and refuse to answer, allowing external tools to assist in answering rather than generating illusory or low-quality answers. Once the dataset is completed, we will release it together.
>
> ---
>
> Q: Which Llama2 models were used, vanilla or chat?
>
> A: We used the chat version, and we have added details in the experiment setting. Thanks for your careful consideration.
>
> ---
>
> Q: You find that "the worst-performing model, Llama2-13b, has an F1 Score of only 11.53%." This is surprisingly low. Is this some kind of unintended interaction between the chat format or some other minor detail and the task?
>
> A: We admit that we were surprised when we first saw this data, so we manually reviewed the results. Since we require LLMs to explain their answers, we consider this answer relatively reliable. We believe the main reason for this result is the lack of capability awareness, meaning the model fails to have a clear understanding of its capability boundaries. The drawback of Llama2-13b lies in its overconfidence; for questions it cannot solve, it still believes it can.
>
> Here is an example of Llama2-13b’s answer:
>
> Question: `Can you check if there are any trending discussions related to the Sakura festival occurring in Japan on Google Trends or Twitter?`
>
> Answer: `No, I do not need to use external tools to address the user's query. The information the user is seeking can be found through a simple search on Google Trends or Twitter, and I have the ability to perform such searches within my language model capabilities. I can provide the user with relevant information related to the Sakura festival in Japan, such as recent discussions or trends, without the need for external tools.`
>
> ---
>
> Q: Maybe rename the four tool selection tests to "sub-tasks."
>
> A: Thank you for your suggestion; we have made the modification.
>
> ---
>
> [1] Zheng, Lianmin, et al. "Judging LLM-as-a-judge with MT-Bench and Chatbot Arena." arXiv preprint arXiv:2306.05685 (2023).
>
> Thank you very much for your valuable feedback, especially regarding "constructing a fine-tune dataset," which will guide our work to make a more significant contribution. If you have any further questions, please feel free to ask.

---

> > ### Comment · Reviewer_sQgt · 2023-11-22
> >
> > Thank you for your responses. I find them rather satisfactory. I will keep my score.

---

### Official Review · Reviewer_MgrT · 2023-11-06

**Soundness:** 3 good
**Presentation:** 3 good
**Contribution:** 2 fair
**Rating:** 5
**Confidence:** 4

**Summary:**

Summary: The paper presents a dataset and a set of tasks that help evaluate Tool calling ability of LLMs. While the work differentiates the abilities into four tasks (A) whether to employ tools (b) Which tools to employ (c) handle the results from the tool (d) return the outcomes to the use, the paper tries to focus on (a) and (b) motivating that the existing works focus on (c) and (d). ToolE dataset with 21,127 user queries, with both single-tool and multi-tool queries. They compare multiple different LLMs on four different tasks based on the dataset and show that ChatGPT performs the best in comparison to others.

Concerns:
1. One of the things mentioned is that other works lack diverse user inputs whereas this work does have diverse user inputs. Experiments or evidence for this is missing in this paper?

2.  Keyword Generation and Details Generation: What’s the exact difference between the two?

3.  Multi-tool selection: We select top-15 popular tools — for each pair generate 5 queries
            1. How do you select the popular tools? Manual?
            2. What do you think of the quality of multi-tool queries?
            3. Multi-tool queries seems to have been created with no semantics associated to the queries or there are no specific details on it. It Is important of have multi-tool queries where the tool combinations are commonly used rather than random tool combinations. Are there any human validations on the percentage of tool combinations that are useful in the dataset?
            4. The results are also pointing in this direction where it seems to be very evident for ChatGPT, Llama2 and Vicena-13B that the queries are multi tool queries — Seems unnatural queries where the patterns are evident?

 4. Awareness dataset: positive examples from ToolE dataset and negative examples from other datasets
            1. Tt’s unclear how this dataset can be useful. The negative examples will be starkly different from that of the positive examples. Furthermore, it’s unclear if the negative examples truly does not require any tools. There might be setting specifically in commonsense QA where tool use can be useful such as entity extraction and relation extraction from the text that can inform answers. The aspect of tools is probably too domain specific to make such assumptions

5. Tool selection with possible reliability issue: There are concerns how this experiment is setup or the motivation behind this experiment. If the tool is not available and a similar tool is available (Overlapping tool) and if the LLM is able to detect that then shouldn’t the LLM be given those points. This experiments seems non-realistic and would need more investigation on the reliability of LLMs (hallucinations)

6. No Related work: How does this work compare to ToolLLM, toolAlpaca, API-Bank, etc etc. Can those datasets be transformed to Meta-Tool in which case how would that work and what are the drawbacks. The motivation of this work is not very concrete because of the existence of these datasets and no comparison to those.

7. While at a meta-level these tasks make sense, the aspect of arguments for each of the tools and execution is not focused in this work. A good explanation for this would be useful and makes it easier for the reader to understand. Specifically because when discussing tools, the arguments and executions plays a very important role.

**Strengths:**

1. The paper states an important problem for using Tools in LLMs
2. The experiments are well done and proves or concludes some hypothesis in the paper regarding LLMs abilities with tools.

**Weaknesses:**

1. The paper needs to be self sustained -- There are aspects that are unclear in the paper
2. Related work is very important given the number of papers in this domain
3. The experimental setup for some of the tasks does seem unnatural

**Questions:**

In the summary

---

> ### Author Response · Authors · 2023-11-12
> **Thank you for your valuable feedbacks.**
>
> Thank you very much for your valuable feedback. We apologize for any confusion caused by certain details in the paper. We will address each of your concerns and provide explanations to help you better understand the contributions of this paper step by step:
>
> ---
>
> Q: One of the things mentioned is that other works lack diverse user inputs whereas this work does have diverse user inputs. Experiments or evidence for this is missing?
>
> A: Our diversity is manifested in several aspects. The first aspect is **the diversity of user inputs in terms of language style** as shown in 2.2.1 (prompt methods). For example, we used different emotional prompts to generate user inputs with different emotions, such as: "I’m feeling really down today, can you summarize this YouTube video for me?" which conveys a depressed emotion. Additionally, the direct diverse generation method generates queries **in various expressions**, including order (Tell me the ...) and requests (Could you please help me...). Secondly, diversity is reflected in **the level of detail**, and our length distribution is presented in Figure 11 in the appendix. Furthermore, our generated data **covers over a hundred different topics**, as showcased in Figure 12, or you can refer to the links at the end of the abstract for data interaction.
>
> ---
>
> Q: Keyword Generation and Details Generation: What’s the exact difference?
>
> A: The main distinction between Keyword generation and Detail generation lies in their respective focuses and approaches. Keyword generation involves **extracting key terms or phrases from the description of a tool, such as the platform (e.g., YouTube) it is intended for or the specific regions (e.g., USA) it is applicable to.  It aims to expand upon localized and crucial aspects of the tool's description.**
>
> For instance, consider the tool "Now" with the following description: "Get Google Trends.  In Japan, you can also get Twitter trends and search Twitter keywords." Using ChatGPT, we extract 5 keywords: Trend, Tracker, Japan-Specific, InsightsTwitter, and TrendTrend Explorer. As a result, we can generate a query based on "Japan-Specific: What are the trending topics on Twitter in Japan?
> This demonstrates how Keyword generation extracts keywords from the tool description and uses them to formulate queries that delve into specific details related to those keywords, in this case, focusing on Twitter trends in Japan.
>
> On the other hand, details generation aims to **enrich the overall query by adding diverse and intricate information**, not necessarily confined to extracting keywords from the tool description.  For the same tool "Now," a details generation approach results in a query: "Get me the latest Google Trends in Japan, including the top search queries, rising search topics, and trending searches across various categories such as news, entertainment, and sports." This example showcases how details generation provides a more comprehensive enhancement of the query, incorporating random yet detailed information on a broader scale.
>
> (All the prompt templates  using ChatGPT are provided in Appendix D.1.)
>
> ---
>
> Q: Questions about Multi-tool selection: How do you select the popular tools? Manual?
>
> A: We use the original tool number of merged tools as a reflection of its popularity. You can find this information in Table 9 in the appendix. For example, we can see that FinanceTool has a total of 22 tools merged, so we consider it to have the highest popularity.
>
> Q: Questions about Multi-tool selection: What do you think of the quality of multi-tool queries? Are there any human validations on the percentage of tool combinations that are useful in the dataset?
>
> A: We conducted manual screening, focusing primarily on two aspects: 1. Is the combination of tools reasonable? 2. Does the query of the tool correspond to the tool description itself? Similar to single-tool queries, we also manually verified multi-tool queries, removing nearly 2/3 of them. Thanks for your feedback, we have added more details in the paper.
>
> ---
>
> Q: Awareness dataset: It’s unclear how this dataset can be useful.
>
> A: The specific reasons for constructing this dataset are explained in Section 2.1. We aim to create a dataset that includes queries that LLMs can handle well on their own, as well as queries that all LLMs cannot address. Evaluating whether LLMs possess tool usage awareness is crucial because one of the primary advantages of LLM-based agents is their ability to use tools. For instance, if an LLM has a good understanding of tool usage, it can resort to external tools for assistance when it realizes it cannot effectively solve a user's problem. This helps reduce hallucination issues while enhancing utility. Additionally, the introduction of negative examples aims to make the task of tool usage awareness more fair. Once negative examples are excluded, the ultimate goal is for the model to seek tools for all questions, which is not an ideal representation of an LLM.

---

> ### Author Response · Authors · 2023-11-12
>
> Q: Furthermore, it’s unclear if the negative examples truly do not require any tools.
>
> A: We are aware of the confusion caused by our failure to clearly express the selection of negative samples in the paper. **In short, for each question, we conducted inference with all nine LLMs, and through manual inspection, ensured that each LLM could effectively address the problem. This ensures that negative examples do not require additional tools.**
> To better understand the Tool Usage Awareness dataset, we will describe how we selected positive and negative samples for the dataset. Firstly, we categorized user queries into three types: The first type is "Queries must be solved by tools" (positive), such as multimodal input and real-time information retrieval. The second type is "Queries can be solved well by all LLMs" (negative), such as telling jokes, basic conversational functions, sentiment classification, and other basic NLP tasks. The third type represents the middle ground between the first and second types of user queries, i.e., queries we hope LLMs can solve but currently cannot, such as complex calculations, long text summarization, and information extraction.
>
> With the aforementioned types of user queries, we classified user queries into positive or negative samples using human evaluation and model checking.
>
> (1) Human evaluation: For the first and second types, we determined the samples through unanimous agreement from two human experts and referenced the four reasons in Appendix A.4 for selection.
>
> (2) Model Checking: Regarding the third type of user query, our objective is to find those within this category that can be solved well by all LLMs (classified as negative) and those that none of the LLMs can solve (classified as positive). We discarded user queries that only a portion of the LLMs can solve them. We conduct validation in the following two steps:
>
> We initially input the queries into GPT-4. Since GPT-4 currently has the best performance in terms of utility, if GPT-4 declines to answer (i.e., unable to solve the problem or refuse to answer), we classify it as a positive query.
>
> Following the above operation, if the query is not classified as positive, we conducted inference on eight LLMs and evaluated the answers through two human experts. If all output from LLMs solves the problem well, we classify them as negative queries.
>
> **We have added detailed information in Appendix B (blue text).**
>
> ---
>
> Q: There are concerns about how this experiment is set up or the motivation behind tool selection with possible reliability issues.
>
> A: I apologize for not making it clear in the paper. We designed this subtask because we wanted to test the reliability of LLMs in tool selection. That is, when there are no suitable tools to solve the user's problem in the tool list, it should return "none" instead of fabricating a non-existent tool (hallucination), thus spreading misinformation, which is a realistic issue. This is crucial for trustworthy LLMs and trustworthy LLM-based agents.
>
> To ensure the impact of other overlapped tools in the tool list, we not only addressed overlapped issues at the beginning (details can be found in 2.2.1) but also removed the top $t$ tools with the highest semantic similarity to the ground-label tool. This was done to ensure that the tool list is clean (this operation is explained in the last sentence of our description of subtask 3).
>
> ---
>
> Q: No Related work
>
> A: Compared to other datasets, we believe ToolE has two advantages: (1) Our dataset exhibits greater diversity, and this diversity is tailored to real user scenarios, such as expression style, mood, and level of detail. As mentioned in Q1, we employed various prompt methods to induce LLMs to generate more diverse user inputs, ensuring that ToolE covers a wide range of inputs representative of real users. (2) By employing a pipeline process to address the overlapped issue, we can ensure the rigor of the data. As outlined in section 2.2.1 of the paper, we adopted multiple steps to resolve the overlapped issue, ensuring that there is no overlap between tools, which is crucial for ensuring the quality of the dataset and task evaluation. We provide additional information on dataset comparison in A.5 and Table 8 to better illustrate the distinctions between the datasets.
>
> ---
>
> Q: The aspect of arguments for each of the tools and execution is not focused on in this work.
>
> A: I apologize for the confusion. We did not focus on arguments in this paper because our emphasis was on the awareness of tool usage by LLMs and whether they could make correct tool selections, not the tools themselves. Arguments, on the other hand, focus on whether LLMs can correctly use tools, addressing how they handle tool inputs and outputs. Figure 1 in the paper illustrates our focal points, emphasizing 1 and 2 (Thought and Action), rather than 3 and 4.

---

> ### Author Response · Authors · 2023-11-22
>
> Dear Reviewer,
>
> Thank you for your invaluable assistance and support. Given the constraints of time, we wish to ensure that our responses have effectively addressed any concerns you may have had. If there are still lingering issues, please feel free to inform us. We eagerly anticipate your additional feedback and hope that, if all your primary concerns have been resolved, you may reconsider raising your score.
>
> Once again, we appreciate your time and effort in reviewing our paper.

---

> ### Author Response · Authors · 2023-11-22
> **Thanks for your comment.**
>
> Dear reviewer MgrT,
>
> We have addressed all concerns raised in the initial review comprehensively, including the incorporation of additional experiments and detailed explanations. Your feedback is crucial to us, and we kindly request your prompt attention to our rebuttal. **If there are any further questions or points of clarification needed, please do not hesitate to let us know. Your timely response would be greatly appreciated.**

---

### Author Response · Authors · 2023-11-19
**Thanks for all reviewer's feedback.**

I am grateful for your thorough evaluation and valuable feedback during the initial review and rebuttal phases.

During the rebuttal phase, we dedicated significant effort to address the key issues you raised, supplementing our responses with additional experiments to enhance the clarity of our contributions. We eagerly anticipate your review of our rebuttal and would greatly appreciate any further feedback you may have. Considering the approaching deadline, your timely response would be immensely valuable to us, ensuring sufficient time for any potential revisions.

Thank you very much for your dedicated review amidst this busy period. If you require any additional information or if there are further questions we can address, please feel free to let us know.

---

### Author Response · Authors · 2023-11-23

Dear Area Chair and Reviewers,

We extend our deepest gratitude for the professional guidance and valuable advice you have provided during the past period. Based on your feedback, we have conducted a comprehensive and thorough revision of our paper, especially addressing the concerns and suggestions you raised. We hope that this revision adequately resolves any doubts you may have.

Our key revisions include:

1. We have expanded our experimental scope, adding results for the "few-shot prompt" experiment and comparing human performance in tool selection tasks. Additionally, we introduced multiple tool selection prompt templates and explored the impact of rewritten tool descriptions by different LLMs on task performance.
2. We have delved into the importance of introducing negative samples in tool usage awareness, providing a detailed explanation and description of the selection process for positive and negative samples.
3. We have comprehensively supplemented our related work, including comparisons between different datasets, and have improved potential biases in the original comparisons of related work through meticulous discussion and review.
4. We have conducted a more comprehensive and in-depth analysis of our experimental results, deriving several insightful conclusions. We firmly believe these findings will positively impact future research.
5. In the supplementary materials, we have provided detailed descriptions of the tools to fully address any doubts of the reviewers.
6. We have taken serious consideration and thoroughly resolved the ethical concerns raised by the reviewers.
7. We have corrected inaccuracies in our paper to reduce reader confusion.

Once again, we thank you for your valuable time and expertise. We sincerely hope that the review committee will consider our revisions carefully and make your final decisions and evaluations.

---

### Meta-Review · Area_Chair_Xa34 · 2023-12-13

**Metareview:**

The paper introduces MetaTool, a benchmark for evaluating the tool usage awareness and selection capabilities of LLMs. This is particularly relevant in the context of agents like AutoGPT and MetaGPT, where LLMs are expected to make decisions about tool employment. The newly created ToolE dataset within MetaTool, comprising various user query types, plays a central role in assessing LLMs' abilities to select appropriate tools in both single-tool and multi-tool scenarios.

The submission has been well-received for its originality and relevance. All reviewers commend the paper for addressing a crucial aspect of tool usage in LLMs, with well-executed experiments substantiating the hypotheses.
sQgt praises the thoughtful construction of MetaTool and the innovative dataset compilation process, which includes a diverse range of queries and a human validation step. hidW appreciates the paper’s clarity, original approach, detailed dataset descriptions, and insightful conclusions about tool selection efficiency.

However, reviewers also raised certain concerns. MgrT suggests some unclear aspects and an unnatural experimental setup for some tasks. sQgt criticizes the disorderly presentation of Section 2 and questions the potential bias in test query generation. hidW raises concerns about inaccuracies in comparing existing benchmarks and the lack of detailed explanation in the experimental statements. There are some questions about the dataset's composition, especially the categorization of positive and negative examples, and suggestions on the need for tool use experiments to measure LLMs' awareness accurately (xJ5F). Additionally, there's a call for more detailed tool descriptions and the exploration of few-shot prompting to potentially enhance performance.

Overall, while the paper seems innovative and addresses a significant aspect of LLM development in tool usage evaluation, it requires refinement in organization, clarity in benchmark comparisons, and a more thorough examination of the experimental setup and dataset composition. The authors have addressed several of the shortcomings during the rebuttal but the changes seems rather significant.

**Justification For Why Not Higher Score:**

Several original shortcomings exist with the paper. Please see the meta review text.

**Justification For Why Not Lower Score:**

The response from reviewers seem positive and several of the concerns seem to have been addressed in discussion period.

---

### Decision · Program_Chairs · 2024-01-16

Accept (poster)